# Hyper-Decision Transformer for Efficient Online Policy Adaptation

**Mengdi Xu**[1], **Yuchen Lu**[2], **Yikang Shen**[3], **Shun Zhang**[3], **Ding Zhao**[1] **& Chuang Gan**[3,4]
[1] Carnegie Mellon University, [2] University of Montreal, Mila,
[3] MIT-IBM Watson AI Lab, [4] UMass Amherst

## ABSTRACT

Decision Transformers (DT) have demonstrated strong performances in offline reinforcement learning settings, but quickly adapting to unseen novel tasks remains challenging. To address this challenge, we propose a new framework, called Hyper-Decision Transformer (HDT), that can generalize to novel tasks from a handful of demonstrations in a data- and parameter-efficient manner. To achieve such a goal, we propose to augment the base DT with an adaptation module, whose parameters are initialized by a hyper-network. When encountering unseen tasks, the hyper-network takes a handful of demonstrations as inputs and initializes the adaptation module accordingly. This initialization enables HDT to efficiently adapt to novel tasks by only fine-tuning the adaptation module. We validate HDT's generalization capability on object manipulation tasks. We find that with *a single expert demonstration and fine-tuning only* $0.5\%$ *of DT parameters*, HDT adapts faster to unseen tasks than fine-tuning the whole DT model. Finally, we explore a more challenging setting where expert actions are not available, and we show that HDT outperforms state-of-the-art baselines in terms of task success rates by a large margin. Demos are available on our project page.[1]

## 1 INTRODUCTION

Building an autonomous agent capable of generalizing to novel tasks has been a longstanding goal of artificial intelligence. Recently, large transformer models have shown strong generalization capability on language understanding when fine-tuned with limited data (Brown et al., 2020; Wei et al., 2021). Such success motivates researchers to apply transformer models to the regime of offline reinforcement learning (RL) (Chen et al., 2021; Janner et al., 2021). By scaling up the model size and leveraging large offline datasets from diverse training tasks, transformer models have shown to be generalist agents successfully solving multiple games with a single set of parameters (Reed et al., 2022; Lee et al., 2022). Despite the superior performance in the training set of tasks, directly deploying these pre-trained agents to novel unseen tasks would still lead to suboptimal behaviors.

One solution is to leverage the handful of expert demonstrations from the unseen tasks to help policy adaptation, and this has been studied in the context of *meta imitation learning* (meta-IL) (Duan et al., 2017; Reed et al., 2022; Lee et al., 2022). In order to deal with the discrepancies between the training and testing tasks, these works focus on fine-tuning the whole policy model with either expert demonstrations or online rollouts from the test environments. However, with the advent of large pre-trained transformers, it is computationally expensive to fine-tune the whole models, and it is unclear how to perform policy adaptation efficiently (Figure 1 (a)). We aim to fill this gap in this work by proposing a more parameter-efficient solution.

Moreover, previous work falls short in a more challenging yet realistic setting where the target tasks only provide demonstrations without expert actions. This is similar to the state-only imitation learning or Learning-from-Observation (LfO) settings (Torabi et al., 2019; Radosavovic et al., 2021), where expert actions are unavailable, and therefore we term this setting as *meta Learning-from-Observation* (meta-LfO). As a result, we aim to develop a more general method that can address both meta-IL and meta-LfO settings.

---

[1]Project Page: https://sites.google.com/view/hdtforiclr2023/home.

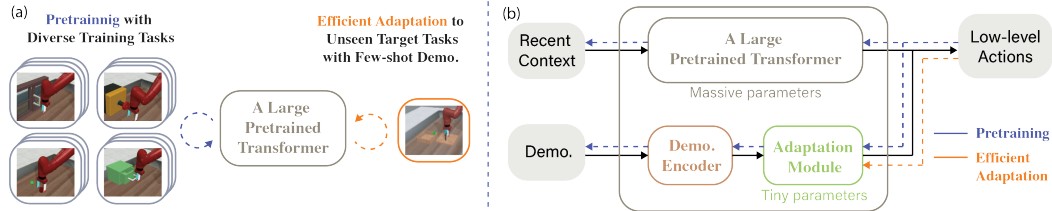

Figure 1: Efficient online policy adaptation of pre-trained transformer models with few-shot demonstrations. To facilitate data efficiency, we introduce a demonstration-conditioned adaptation module that helps leverage prior knowledge in the demonstration and guide exploration. When adapting to novel tasks, we only fine-tune the adaptation module to maintain parameter efficiency.

The closest work to ours is Prompt-DT (Xu et al., 2022), which proposes to condition the model behavior in new environments on a few demonstrations as prompts. While the method is originally evaluated for meta-IL, the flexibility of the prompt design also allows this method to be useful for meta-LfO. However, we find that Prompt-DT hardly generalizes to novel environments (as is shown empirically in Section 4), since the performance of the in-context learning paradigm, e.g., prompting, is generally inferior to fine-tuning methods (Brown et al., 2020). The lack of an efficient adaptation mechanism also makes Prompt-DT vulnerable to unexpected failures in unseen tasks.

In order to achieve the strong performance of fine-tuning-based methods as well as maintain the efficiency of in-context learning methods, we propose Hyper-Decision Transformer (HDT) for large pre-trained Decision Transformers (DT) (Chen et al., 2021). HDT composes of three key components: (1) a multi-task pre-trained DT model, (2) adapter layers that can be updated when solving novel unseen tasks, and (3) a hyper-network that outputs parameter initialization of the adapter layer based on demonstrations. The pre-trained DT model encodes shared information across diverse training tasks and serves as a base policy. To mimic the performance of fine-tuning methods, we introduce an adapter layer with a bottleneck structure to each transformer block (Houlsby et al., 2019). The parameters in the adapter layers can be updated to adapt to a new task. Moreover, it only adds a small fraction of parameters to the base DT model. The adapter parameters are initialized by a single hyper-network conditioning on the demonstrations with or without actions. In such a way, the hyper-network extracts the task-specific information from demonstrations, which is encoded into the adapter's initialized parameters.

We evaluate HDT in both meta-IL (with actions) and meta-LfO (without actions) settings. In meta-IL, the adapter module could directly fine-tune in a supervised manner as few-shot imitation learning. In meta-LfO, the agent interacts with the environment and performs RL, while conditioning on the expert states. We conduct extensive experiments in the Meta-World benchmark Yu et al. (2020), which contains diverse manipulation tasks requiring fine-grind gripper control. We train HDT with 45 tasks and test its generalization capability in 5 testing tasks with unseen objects, or seen objects with different reward functions. Our experiment results show that HDT demonstrates strong data and parameter efficiency when adapting to novel tasks.

We list our contributions as follows:

1. We propose Hyper-Decision Transformer (HDT), a transformer-based model which generalizes to novel unseen tasks maintaining strong data and parameter efficiency.

2. In the meta-IL setting with *only one expert demonstration*, HDT only fine-tunes a small fraction (0.5%) of the model parameters and adapts faster than baselines that fine-tune the whole model, demonstrating strong parameter efficiency.

3. In the meta-LfO setting with only 20-80 online rollouts, HDT can sample successful episodes and therefore outperforms baselines by a large margin in terms of success rates, demonstrating strong data efficiency.

## 2 RELATED WORK

**Transformers in Policy Learning.** Transformer (Vaswani et al., 2017) has shown success in natural language tasks thanks to its strong sequence-modeling capacity. As generating a policy in RL is

essentially a sequence prediction problem, Chen et al. (2021) propose Decision Transformer (DT) that adapts the Transformer architecture to solve offline RL problems, achieving performances comparable to traditional offline RL algorithms. This work also inspires adaptions of Transformer in different RL settings. Zheng et al. (2022) propose Online DT, which enables DT to explore online and applies it in an online RL setting. Xu et al. (2022) use the prompt in Transformer to provide task information, which makes the DT capable of achieving good performance in different tasks. Reed et al. (2022) also exploit the prompt in Transformer and train one Transformer model to solve different games or tasks with different modalities. Lee et al. (2022) propose multi-game DT which shows strong performance in the Atari environment (Bellemare et al., 2013). Furuta et al. (2021) identify that DT solves a hindsight information matching problem and proposed Generalized DT that can solve any hindsight information matching problem.

**One-shot and Meta Imitation Learning.** Traditional imitation learning considers a single-task setting, where the expert trajectories are available for a task and the agent aims the learn an optimal policy for the same task. One-shot and meta-imitation learning aims to improve the sample efficiency of imitation learning by learning from a set of training tasks and requires the agent to generalize to different tasks. Duan et al. (2017) train a one-shot imitator that can generate optimal policies for different tasks given the demonstrations of the corresponding task. Finn et al. (2017b) enable a robot to learn new skills using only one demonstration with pixel-level inputs. James et al. (2018) first learn task embeddings, and use the task embedding of the test task jointly with test task demonstrations to find the optimal control policy in robotic tasks.

**Learning from Observations.** When deployed in a new environment or tested on a new task, it may be challenging for an agent to achieve good performance without any information about the task. Therefore, the agent can benefit from starting with imitating some expert trajectories (Argall et al., 2009). Usually, expert trajectories consist of a sequence of state, action pairs rolled out from an expert policy. However, assuming the availability of both states and actions can prevent the agent from learning from many resources (for example, online video demonstrations). So we consider a setting where such trajectories may only contain states and rewards rolled out from the expert policies. This is considered as learning from observations in the literature (Torabi et al., 2019). Prior work trains an inverse dynamics model and uses it to predict actions based on states (Radosavovic et al., 2021) and enhances learning from observation methods by minimizing its gap to learning from demonstrations (Yang et al., 2019). Transformers are also applied in this problem to achieve one-shot visual imitation learning (Dasari & Gupta, 2020).

**Hyper-networks.** The idea of using one network (the hyper-network) to generate weights for another (the target network) is first introduced in Ha et al. (2016). In the continuing learning setting, Von Oswald et al. (2019) proposes generating the entire weights of a target network while conditioned on the task information. In the robotic setting, Xian et al. (2021) considers a model-based setting and uses hyper-networks to generate the parameters of a neural dynamics model according to the agent's interactions with the environment.

**Parameter-efficient Adaptation.** When modern neural network models become increasingly large, generating or fine-tuning the full model becomes increasingly expensive. Lester et al. (2021) introduces "soft prompts" to alternate the output of a pre-trained transformer model. Task-specific soft prompts enable the transformer model to output answers for the given task. Liu et al. (2022) proposes adding a few controlling vectors in the feedforward and the attention blocks to control the behavior of a pre-trained transformer model. Mahabadi et al. (2021) uses a hyper-network to generate task-specific adaptation modules for transformer models from a task embedding. This method enables efficient adaptation by just learning the task embedding vector.

## 3    HYPER DECISION TRANSFORMER

To facilitate the adaptation of large-scale transformer agents, we propose Hyper-Decision Transformer (HDT), a Transformer-based architecture maintaining data and parameter efficiency during adaptation to novel unseen tasks. HDT consists of three key modules as in Figure 2: a base DT model encoding shared knowledge across multiple tasks (Section 3.2), a hyper-network representing a task-specific meta-learner (Section 3.4), and an adaptation module updated to solve downstream unseen tasks (Section 3.3). After presenting the model details, we summarize the training and adaptation algorithms in Section 3.5.

### 3.1 PROBLEM FORMULATION: EFFICIENT ADAPTATION FROM OBSERVATIONS

We aim to improve the generalization capability of large transformer-based agents with limited online interactions and computation budgets. Existing large transformer-based agents have demonstrated strong performances in training tasks, and may have sub-optimal performances when directly deployed in novel tasks (Reed et al., 2022; Lee et al., 2022; Xu et al., 2022). Although the discrepancy between the training and testing tasks has been widely studied in meta RL (Finn et al., 2017a; Duan et al., 2016), efficient adaptations of large transformer-based agents are still non-trivial. Beyond the data efficiency measured by the number of online rollouts, we further emphasize the parameter efficiency measured by the number of parameters updated during online adaptation.

We formulate the environment as an Markov Decision Process (MDP) represented by a 5-tuple $\mathcal{M} = (\mathcal{S}, \mathcal{A}, P, R, \mu)$. $\mathcal{S}$ and $\mathcal{A}$ are the state and the action space, respectively. $P$ is the transition probability and $P : \mathcal{S} \times \mathcal{A} \times \mathcal{S} \to \mathbb{R}$. $R$ is the reward function where $R : \mathcal{S} \to \mathbb{R}$. $\mu$ is the initial state distribution. Following Xu et al. (2022); Torabi et al. (2019), we consider efficient adaption from demonstrations with a base policy pre-trained with a set of diverse tasks. Formally, we assume access to a set of training tasks $\boldsymbol{T}^{train}$. Each task $\mathcal{T}_i \in \boldsymbol{T}^{train}$ is accompanied by a large offline dataset $\mathcal{D}_i$ and a few demonstrations $\mathcal{P}_i$, where $i \in [\|\boldsymbol{T}^{train}\|]$. The dataset $\mathcal{D}_i$ contains trajectories $h$ collected with an (unknown) behavior policy, where $h = (s_0, a_0, r_0, \cdots, s_H, a_H, r_H)$ with $s_i \in \mathcal{S}$, $a_i \in \mathcal{A}$, $r$ as the per-timestep reward, and $H$ as the episode length.

We consider the efficient adaptation problem in two settings: meta-imitation learning (meta-IL) and meta-learning from observations (meta-LfO). In meta-IL, the demonstration dataset $\mathcal{P}$ consists of full trajectories with expert actions, $\hat{h} = (s_0, a_0, r_0, \cdots, s_H, a_H, r_H)$. In contrast, in meta-LfO, the demonstration dataset $\mathcal{P}$ consists of sequences of state and reward pairs, $h^o = (s_0, r_0, \cdots, s_H, r_H)$. We evaluate the generalization capability in a set of testing tasks $\boldsymbol{T}^{test}$, each accompanied with a demonstration dataset $\mathcal{P}$. Note that the testing tasks are not identical to the training tasks and may not follow the same task distribution as the training ones. For instance, in manipulation tasks, the agent may need to interact with unseen objects.

### 3.2 DECISION TRANSFORMER (DT) AS THE PRE-TRAINED AGENT

Large transformer-based agents are highly capable of learning multiple tasks (Lee et al., 2022; Reed et al., 2022). They follow offline RL settings and naturally cast RL as sequence modeling problems (Chen et al., 2021; Janner et al., 2021). By tasking a historical context as input, transformers could leverage more adequate information than the observations in the current timestep, especially when the dataset is diverse and the states are partially observable. In this work, we follow the formulation of DT (Chen et al., 2021), which autoregressively generates actions based on recent contexts. At each timestep $t$, DT takes the most recent $K$-step trajectory segment $\tau$ as input, which contains states $s$, actions $a$, and rewards-to-go $\hat{r} = \sum_{i=t}^{T} r_i$.

$$\tau = (\hat{r}_{t-K+1}, s_{t-K+1}, a_{t-K+1}, \ldots, \hat{r}_t, s_t, a_t). \tag{1}$$

DT predicts actions at heads, where action $a_t$ is the action the agent predicts in state $s_t$. The model minimizes the mean squared error loss between the action predictions and the action targets. In contrast to behavior cloning (BC) methods, the action predictions in DT additionally condition on rewards-to-go and timesteps. The recent context contains both per-step and sequential interactions with environments, and thus encodes task-specific information. Indeed, in Lee et al. (2022), recent contexts could embed sufficient information for solving multiple games in the Atari environment (Bellemare et al., 2013). While in situations where a single state has different optimal actions for different tasks, additional task-specific information (eg. the demonstration in target tasks) injected into the model further helps improve the multi-task learning performance (Xu et al., 2022).

In this work, we pre-train a DT model with datasets from multiple tasks as the base agent. It is worth noting that the base DT model could be replaced by any other pre-trained transformer agent.

### 3.3 ADAPTATION MODULE

To enable parameter-efficient model adaptation, we insert task-specific adapter layers to transformer blocks. Adapter-based fine-tuning is part of parameter-efficient fine-tuning of large language models and generalizes well in NLP benchmarks (Mahabadi et al., 2021). The adapter layer only contains

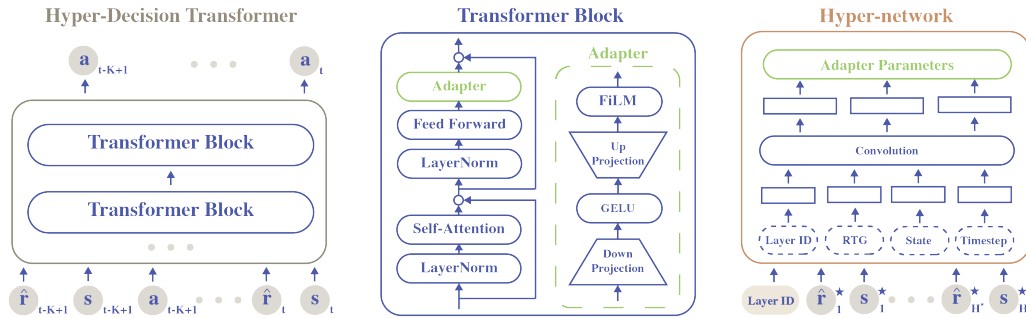

Figure 2: Model architecture of Hyper-Decision Transformer (HDT). Similar to DT, HDT takes recent contexts as input and outputs fine-grind actions. To encode task-specific information, HDT injects adapter layers into each decoder block. The adapter layer's parameters come from a stand-alone hyper-network that takes both demonstrations without actions and the decoder's layer id.

a small number of parameters compared with the base DT policy. When fine-tuning on the downstream unseen tasks, HDT only updates the adapter layers with the base DT model's parameters frozen. Such a task-wise adaptation mechanism helps maintain parameter efficiency and avoid negative interference where adapting to a new task could lead to decreasing performances in other tasks.

Considering that DT is a decoder-only transformer model, we insert one adapter module into each decoder block and place it before the final residual connection, as shown in Figure 2 (middle). Let the hidden state be $\boldsymbol{X} \in \mathbb{R}^{K \times d_x}$, where $K$ is the context length and $d_x$ is the embedding dimension for each token. For each decoder block with layer id $l$, its adapter layer contains a down-projection layer $\boldsymbol{D}_l \in \mathbb{R}^{d_x \times d_h}, d_h < d_x$, a GELU nonlinearity, an up-projection layer $\boldsymbol{U}_l \in \mathbb{R}^{d_h \times d_x}$, and a feature-wise linear modulation (FiLM) layer $FiLM_l$ (Perez et al., 2018). The bottleneck architecture constructed by the down- and up-projection layers helps reduce the adapter's parameters. The FiLM layer consists of weights $\gamma_l \in \mathbb{R}^{d_x}$ and feature-wise biases $\beta_l \in \mathbb{R}^{d_x}$, which help selectively manipulate features. Formally, the adapter layer transforms the hidden state $\boldsymbol{X}$ by

$$\text{Adapter}_l(\boldsymbol{X}) = (GELU(\boldsymbol{X}\boldsymbol{D}_l)\boldsymbol{U}_l) \odot \gamma_l + \beta_l, \tag{2}$$

where $\odot$ is the Hadamard product. For each task, we initialize the adapter layer with a shared hyper-network across multiple tasks and encoder layers as described in Section 3.4.

## 3.4 HYPER-NETWORK

To promote data-efficient online adaptation, we propose to use a shared hyper-network to generate the adapter layers' initial parameters. The hyper-network captures common information across tasks by multi-task training. It also encodes task-specific information for each task by taking the task demonstration without actions $h^o$ as input, applicable for both meta-IL and meta-LfO settings. Given an unseen task, $\mathcal{T} \in \boldsymbol{T}^{test}$ and its accompanied $h^o$, the trained hyper-network could generate a good parameter initialization that facilitates quick online rollouts. Motivated by Mahabadi et al. (2021), HDT further utilizes a compact hyper-network for generating adapters of each block and takes the layer id $l$ as the additional input.

The hyper-network module consists of linear layers to get embeddings for each modality, a 1D convolutional module $Conv$ to get a shared task-specific encoding $z \in \mathbb{R}^{H \cdot d_z}$, where $H$ is the demonstration length, and a set of linear layers to get adapter parameters. Formally, let the weights of linear embedding layers be $L_s \in \mathbb{R}^{d_s \times d_h}, L_{\hat{r}} \in \mathbb{R}^{1 \times d_h}, L_t \in \mathbb{R}^{1 \times d_h}, L_l \in \mathbb{R}^{1 \times d_h}$ for the state, rewards-to-go, timesteps, and layer id, respectively. Similar to timesteps representing temporal positions, the layer id represents positional information of decoder blocks. Hence, we add layer id embedding to state, action, and reward-to-go tokens. There are four types of parameter-generation layers with weights, $L_D \in \mathbb{R}^{d_x d_h \times d_z}$ for down-projection layers, $L_U \in \mathbb{R}^{d_x d_h \times d_z}$ for up-projection layers, and $L_\gamma \in \mathbb{R}^{d_x \times d_z}, L_\beta \in \mathbb{R}^{d_x \times d_z}$ for FiLM layers. Formally, the hyper-network conducts the following operations.

$$\boldsymbol{U}_l, \boldsymbol{D}_l, \gamma_l, \beta_l = \text{Hyper-network}(h^o, l), \text{ where} \tag{3}$$
$$z = Conv(\text{ concat}(\ L_s(h^o) + L_t(h^o) + L_l(h^o), L_{\hat{r}}(h^o) + L_t(h^o) + L_l(h^o)\ )\ ),$$
$$\boldsymbol{U}_l = L_U z, \ \boldsymbol{D}_l = L_D z, \ \gamma_l = L_\gamma z, \ \beta_l = L_\beta z.$$

---

**Algorithm 1** Hyper-network Training

---

1: **Input:** training tasks $\boldsymbol{T}^{train}$, HDT$_\phi$ with hyper-network parameters $\phi$, training iterations $N$, offline dataset $\mathcal{D} = \{h = (s_0, a_0, r_0, \cdots, s_H, a_H, r_H)\}$, demonstrations $\mathcal{P}$, per-task batch size $M$, learning rate $\alpha_\phi$
2: **for** $n = 1$ **to** $N$ **do**
3:     **for** Each task $\mathcal{T}_i \in \boldsymbol{T}^{train}$ **do**
4:         Sample $M$ trajectory $\tau_i$ of length $K$ from $\mathcal{D}_i$, and a demo. $h_i^o$ from $\mathcal{P}_i$
5:         Get a minibatch $\mathcal{B}_i^M = \{(h_i^o, \tau_{i,m})\}_{m=1}^M$
6:     Get a batch $\mathcal{B} = \{\mathcal{B}_i^M\}_{i=1}^{|\boldsymbol{T}^{train}|}$
7:     $a^{pred} = HDT_\phi(h_i^o, \tau_{i,m}), \forall (h_i^o, \tau_{i,m}) \in \mathcal{B}$
8:     $\phi \leftarrow \phi - \alpha_\phi \nabla_\phi \frac{1}{|\mathcal{B}|} \sum_{(h_i^o, \tau_{i,m}) \in \mathcal{B}} (a - a^{pred})^2$, where $a$ refers to actions in $\tau_{i,m}$

---

**Algorithm 2** Efficient Policy Adaptation without Expert Actions (**meta-LfO**)

---

1: **Input:** testing task $\mathcal{T} \in \boldsymbol{T}^{test}$, HDT$_\psi$ with adapter parameters $\psi$, online rollout budget $N_{epi}$, one-shot demonstration without actions $\mathcal{P}$, batch size $M$, learning rate $\alpha_\psi$, exploration rate $\epsilon$
2: **Initialize** adapter parameters $\psi$ with trained hyper-network
3: **Initialize** empty data buffer $\mathcal{D} = \{\emptyset\}$
4: **while** episode number less than $N_{epi}$ **do**
5:     Collect one episodic trajectory $h$ with $\epsilon$-greedy
6:     Relabel rewards-to-go of $h$ with actual rewards and append $h$ to data buffer $\mathcal{D}$
7:     Sample $M$ segments $\tau$ of length $K$ from $\mathcal{D}$, and a demo. $h^o$ from $\mathcal{P}$
8:     Get a batch $\mathcal{B} = \{(h^o, \tau_m)\}_{m=1}^M$
9:     $a^{pred} \leftarrow HDT_\psi(h^o, \tau_m), \forall (h_i^o, \tau_m) \in \mathcal{B}$
10:     $\psi \leftarrow \psi - \alpha_\psi \nabla_\psi \frac{1}{|\mathcal{B}|} \sum_{(h_i^o, \tau_m) \in \mathcal{B}} (a - a^{pred})^2$, where $a$ refers to actions in $\tau_m$

---

### 3.5 ALGORITHM

**DT Pre-training.** We first pre-train a DT model with datasets across multiple tasks. We assume that the datasets do not contain trajectories collected in testing tasks. We defer the detailed training algorithm in the appendix as in Algorithm 3.

**Training Hyper-network.** We train the hyper-networks as shown in Algorithm 1. For notation simplicity, we denote hyper-network parameters as $\phi$. During hyper-network training, the pre-trained transformer agent's parameters are frozen. HDT only updates hyper-network parameters $\phi$ to make sure the hyper-network copes with the pre-trained agent as well as extracts meaningful task-specific information from demonstrations $h^o$. To stabilize training, each gradient update of $\phi$ is based on a large batch containing trajectory segments in each task. Following Chen et al. (2021), we minimize the mean-squared error loss between action predictions and targets.

**Fine-tuning on Unseen Tasks.** When fine-tuning in unseen target tasks, HDT first initializes adapter parameters (denoted as $\psi$) with the pre-trained hyper-network and then only updates the adapter parameters. When there only exist demonstrations without actions $h^o$ in the target task (**meta-LfO**), HDT initializes an empty data buffer and collects online rollouts with $\epsilon$-greedy as in Algorithm 2. Following Zheng et al. (2022), we relabel the rewards-to-go of the collected trajectories. HDT updates adapters with the same mean-squared error loss as training hyper-networks. When there exist expert actions in the demonstration (**meta-IL**), HDT could omit the online rollout and directly fine-tune the adapter with the expert actions following Algorithm 4 in appendix Section A.

## 4 EXPERIMENTAL SETUP

We conduct extensive experiments to answer the following questions:

- Does HDT adapt to unseen tasks while maintaining parameter and data efficiency?
- How is HDT compared with other prompt-fine-tuning and parameter-efficient fine-tuning methods in the field of policy learning?
- Does the hyper-network successfully encode task-specific information across tasks?
- Does HDT's adaptivity scale with training tasks' diversity, the base DT policy's model size, and the bottleneck dimension?

## 4.1 ENVIRONMENTS AND OFFLINE DATASETS

The Meta-World benchmark (Yu et al., 2020) contains table-top manipulation tasks requiring a Sawyer robot to interact with various objects. With different objects, such as a drawer and a window, the robot needs to manipulate them based on the object's affordance, leading to different reward functions. At each timestep, the Sawyer robot receives a 4-dimensional fine-grind action, representing the 3D position movements of the end effector and the variation of gripper openness. We follow the Meta-World ML45 benchmark. We use a training set containing 45 tasks for pre-training DT and hyper-networks. The testing set consists of 5 tasks involving unseen objects or seen objects but with different reward functions.

For each training task, we collect an offline dataset containing 1000 episodes with the rule-based script policy provided in Yu et al. (2020) and increase the data randomness by adding random noise to action commands. For each testing task, we collect one demonstration trajectory with the script policy. Note that the expert rule-based script policy is tailored to each task and has an average success rate of about 1.0 for each task in the training and testing sets. We set the episode length as 200. Each episode contains states, actions, and dense rewards at each timestep.

## 4.2 BASELINES

We compare our proposed **HDT** with six baselines to answer the questions above. For each method, we measure the *task performance* in terms of the success rate in each testing task, the *parameter efficiency* according to the number of fine-tuned parameters during adaptation, and the *data efficiency* based on the top-2 number of online rollout episodes until a success episode. All methods except for SiMPL utilize a base DT model with size $[512, 4, 8]$, which are the embedding dimension, number of blocks, and number of heads, respectively.

- **PDT.** Prompt Decision Transformer (PDT) (Xu et al., 2022) generates actions based on both the recent context and pre-collected demonstrations in the target task. To ensure a fair comparison, we omit the actions in the demonstration when training PDT with $T^{train}$. During fine-tuning, we conduct prompt-tuning by updating the demonstration with the Adam optimizer. PDT helps reveal the difference between parameter-efficient tuning and prompt-tuning in policy learning.

- **DT.** We fine-tune the whole model parameters of the pre-trained DT model during online adaptation. DT helps show the data- and parameter-efficiency of HDT.

- **HDT-IA3.** To show the effectiveness of the adapter layer in policy learning, we implement another parameter-efficient fine-tuning method, HDT-IA3, motivated by Liu et al. (2022). HDT-IA3 insert weights to rescale the keys and values in self-attention and feed-forward layers. Both HDT-IA3 and HDT do not utilize position-wise rescaling. In other words, HDT-IA3 trains hyper-network to generate rescaling weights shared by all positions in each block. During fine-tuning, only the rescaling weights are updated. We detail HDT-IA3's structure in Section A.3.

- **SiMPL.** Skill-based Meta RL (SiMPL) (Nam et al., 2022) follows SPiRL's setting (Pertsch et al., 2020) and uses online rollouts with RL to fine-tune on downstream unseen tasks. We pick this additional baseline for the meta-LfO, since this setting can be thought of as meta-RL with observations. We aim to show the improvements when having access to expert observations.

- **HDT-Rand.** To reveal the effect of the adapter initialization with pre-trained hyper-networks, we compare HDT with HDT-Rand, which randomly initializes adapter parameters during fine-tuning. HDT-Rand only updates the adapter parameters similar to HDT.

## 5 RESULTS AND DISCUSSIONS

### 5.1 DOES HDT GENERALIZE TO UNSEEN TASKS WITH PARAMETER AND DATA EFFICIENCY?

When fine-tuning in unseen tasks with demonstrations containing no expert actions, HDT achieves significantly better performance than baselines in terms of success rate, parameter efficiency (Table 1), and rollout data efficiency (Table 2).

**Success Rate.** Without fine-tuning, HDT could only achieve a success rate of around 0.12 in testing tasks. In the meta-LfO setting, after collecting a handful of episodes in each task, HDT demonstrates a significant improvement to a success rate of around 0.8, as shown in Table 1. In the door-lock and door-unlock tasks, where the object already shows up in the training tasks, HDT could fully

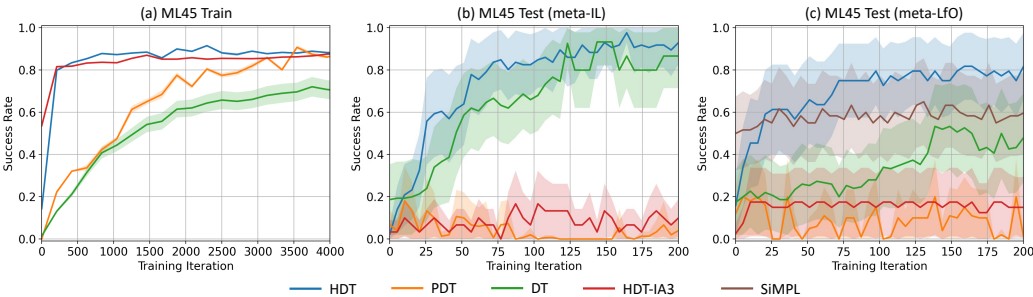

Figure 3: Qualitative results in Meta-World benchmark. Each curve is averaged across 5 seeds. We show the training curves of our proposed **HDT** and baselines in (a), the adaptation performance with a one-shot demonstration containing expert actions (meta-IL) in (b), and adaption performance with a demonstration containing no expert actions (meta-LfO) in (c). When expert actions are unavailable, HDT outperforms baselines by a large margin.

Table 1: Quantitative results on Meta-World ML45 benchmarks.

|        | Model Sizes | | Meta-World ML45 Peformances | | | |
|--------|-------------|------------|-------------|----------------|------------------------|-------------------|
|        | Adaptation  | Percentage | Train       | Test (no-FT)   | Test (meta-IL, 1 shot) | Test (meta-LfO)   |
| HDT    | 69K         | 0.5%       | 0.89± 0.00  | **0.12± 0.01** | **0.93 ± 0.10**        | **0.80 ± 0.16**   |
| PDT    | 6K          | 0.05%      | 0.88± 0.00  | 0.06± 0.05     | 0.04 ± 0.07            | 0.09 ± 0.01       |
| DT     | 13M         | 100%       | 0.70 ± 0.04 | 0.08± 0.03     | 0.87 ± 0.15            | 0.46 ± 0.21       |
| HDT-IA3| 6K          | 0.05%      | 0.88± 0.00  | 0.04± 0.02     | 0.10 ± 0.01            | 0.16 ± 0.14       |

solve the tasks. In more challenging tasks where there exist unseen objects, including bin-picking, box-close, and hand-insert, HDT still outperforms baselines by a large margin, as in Table 2 and Figure 3. SiMPL, without utilizing the demonstrations, could achieve performance improvements while still underperforming our proposed HDT. Such observations show the benefit of leveraging prior information from demonstrations.

**Parameter Efficiency.** In meta-LfO, fine-tuning the whole parameters of the base DT model (denoted as DT) comes third in terms of the average success rate (Figure 3 (c)). However, it requires a significantly larger amount of computation budget to fine-tune 13M parameters compared with our proposed HDT, which fine-tunes 69K parameters (0.05% of 13M) as in Table 1. In the simpler setting with expert actions (meta-IL), all baselines require no online rollouts and update with expert actions. In this case, HDT converges faster than fine-tuning DT as in Figure 3 (b).

**Data Efficiency.** In meta-LfO, we set the online rollout budget as 4000 episodes. As in Table 2, HDT could sample a successful episode in around 20 to 80 episodes, much smaller than the number of episodes required by baselines. Such a comparison shows that the adapter module initialized by the pre-trained hyper-network helps guide the exploration during online rollouts.

## 5.2 ADAPTER LAYERS V.S. OTHER EFFICIENT FINE-TUNING METHODS IN POLICY LEARNING

Prompt-tuning and parameter-efficient fine-tuning are two popular paradigms in tuning pre-trained large language models Liu et al. (2021). We are interested in the effectiveness of methods in both regimes when tuning pre-trained transformer agents. We treat PDT as the representative prompt-tuning baseline and HDT-IA3 as another baseline in the regime of parameter-efficient fine-tuning. We show that HDT outperforms both in terms of adaptation performances in both meta-IL and meta-LfO settings. Although PDT could sample successful episodes quickly during online adaptation (Table 2), purely updating the prompt input hardly improves success rates. We observe similar trends when fine-tuning with expert actions (Figure 3 (b)). HDT-IA3 could sample success episodes but may require a larger number of rollouts than PDT and HDT. As in Figure 3 (c), fine-tuning HDT-IA3 could result in success rate improvements, which is much less than our proposed HDT.

## 5.3 DOES THE HYPER-NETWORK ENCODE TASK-SPECIFIC INFORMATION?

HDT relies on the hyper-network to extract task-specific information and utilizes the information by generating the adapter's parameters. We hope to investigate whether the hyper-network encodes meaningful task-specific details. We visualize the adapter's parameters initialized by the hyper-network for each task in Figure 7. The detailed visualization process is deferred to Section B.2. Figure 7 shows that the adapters' parameters for different tasks could be distinguished from each

Table 2: Per-task Quantitative results without expert actions (**meta-LfO**). For each testing task, we present the success rate averages across episodes and measure the data efficiency based on the average of the smallest and the second smallest numbers of online rollout episodes until collecting a success episode. The symbol "x" means that no successful episodes are sampled during online rollouts. We highlight the best performances.

| | bin-picking | | box-close | | hand-insert | | door-lock | | door-unlock | |
| | success rate | data eff. | success rate | data eff. | success | data eff. | success | data eff. | success | data eff. |
|---|---|---|---|---|---|---|---|---|---|---|
| HDT | **0.60 ± 0.49** | **20** | **0.80 ± 0.34** | 30 | **0.60 ± 0.37** | 80 | **1.00 ± 0.00** | **20** | **1.00 ± 0.00** | **20** |
| PDT | 0.00 ± 0.00 | 40 | 0.06 ± 0.12 | **20** | 0.00 ± 0.00 | **20** | 0.59 ± 0.48 | **20** | 0.00 ± 0.00 | 170 |
| DT | 0.16 ± 0.37 | 1880 | 0.37 ± 0.31 | 1480 | 0.16 ± 0.37 | 640 | 0.38 ± 0.36 | 50 | 0.80 ± 0.40 | **20** |
| HDT-IA3 | 0.00 ± 0.00 | 190 | 0.00 ± 0.00 | 950 | 0.00 ± 0.00 | x | 0.00 ± 0.38 | x | 0.75 ± 0.25 | **20** |
| HDT-Rand | 0.00 ± 0.00 | x | 0.07 ± 0.12 | 280 | 0.00 ± 0.00 | 2600 | 0.42 ± 0.45 | 110 | 0.25 ± 0.36 | 30 |
| HDT-small-train | 0.00 ± 0.00 | 440 | 0.00 ± 0.00 | 440 | 0.83 ± 0.29 | 800 | 0.17 ± 0.29 | 1220 | 0.67 ± 0.58 | **20** |

Figure 4: Ablation results to show the effect of training tasks and model size. Decreasing the adapter's bottleneck hidden size would slow down the convergence when there are expert actions as in (a), and cause a significant performance drop when no expert actions as in (b). Similar trends are observed with decreased base DT's model size as in (c) and (d). With 10 training tasks, HDT-small-train underperforms HDT.

other, showing that the hyper-network indeed extracts task-specific information. To support the argument, we further visualize the environment rollouts in the 5 testing tasks in Section B.1. Compared with HDT-Rand, which randomly initializes adapter layers, HDT has much better adaptation performance with a higher success rate and strong data efficiency (Table 2 and Figure 4).

## 5.4 DOES HDT'S PERFORMANCE SCALE WITH TRAINING TASKS AND MODEL SIZES?

Lee et al. (2022) show that the data diversity and model size are crucial when training large transformer-based models in the multi-game setting. We hope to understand whether similar trends show up when training HDT. The ablation results are summarized in Table 2 and Figure 4. To understand the effect of *training task diversity*, we compare HDT with HDT-small-train, which trains the hyper-network and the base DT model with 10 training tasks. In meta-LfO, HDT-small-train underperforms HDT in terms of success rates and efficiency when sampling successful episodes. Such results show the importance of training task diversity. It is worth noting that with a smaller *DT model size*, the transformer agent could hardly sample successful episodes in meta-LfO and thus no improvements during online adaptation (Figure 4 (d)). The performances also drop with smaller *hidden sizes of the adapter layers*. With expert actions in meta-IL, smaller DT model size and bottleneck dimension lead to slower convergence.

## 6 CONCLUSION

We propose Hyper-Decision Transformer (HDT), a transformer-based agent that generalizes to unseen novel tasks with strong data and parameter efficiency. HDT fine-tunes the adapter layers introduced to each transformer block during fine-tuning, which only occupies 0.5% parameters of the pre-trained transformer agent. We show that in the Meta-World benchmark containing fine-grind manipulation tasks, HDT converges faster than fine-tuning the overall transformer agent with expert actions. Moreover, HDT demonstrates strong data efficiency by initializing adapter layers' parameters with a hyper-network pre-trained with diverse tasks. When expert actions are unavailable, HDT outperforms baselines by a large margin in terms of success rates. We attribute the strong performance to good initializations of the adapter layers, which help HDT achieve successful online rollouts quickly. We hope that this work will motivate future research on how to optimally fine-tune large transformer models to solve downstream novel tasks. Interesting future directions include scaling HDT up to handle embodied AI tasks with high-dimensional egocentric image inputs.

## ACKNOWLEDGMENTS

This project was supported by the DARPA MCS program, MIT-IBM Watson AI Lab, National Science Foundation under grant CAREER CNS-2047454, and gift funding from MERL, Cisco, and Amazon.

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

## A  ADDITIONAL ALGORITHM DESCRIPTIONS

We present the algorithms for pre-training DT models in Section A.1, efficient adaptation of HDT when there exist expert actions in Section A.2, the model architecture of HDT-IA3 in Section A.3.

### A.1  DT PRE-TRAINING

We pre-train a base DT model with large offline datasets collected from training tasks. Each gradient update of the DT model considers information from each training task.

---
**Algorithm 3** DT pre-training
---
1: **Input:** training tasks $\boldsymbol{T}^{train}$, training task size $T^{train} = |\boldsymbol{T}^{train}|$, $DT_\theta$, training iterations $N$, offline dataset $\mathcal{D}$, demonstrations $\mathcal{P}$, per-task batch size $M$, learning rate $\alpha_\theta$
2: **for** $n = 1$ **to** $N$ **do**
3:     **for** Each task $\mathcal{T}_i \in \boldsymbol{T}^{train}$ **do**
4:         Sample $M$ trajectory $\tau_i$ of length $K$ from $\mathcal{D}_i$
5:         Get a minibatch $\mathcal{B}_i^M = \{\tau_{i,m}\}_{m=1}^M$
6:     Get a batch $\mathcal{B} = \{\mathcal{B}_i^M\}_{i=1}^{T^{train}}$
7:     $a^{pred} = DT_\theta(\tau_{i,m}), \forall(\tau_{i,m}) \in \mathcal{B}$
8:     $\mathcal{L}_{MSE} = \frac{1}{|\mathcal{B}|}\sum_{h^{input} \in \mathcal{B}}(a - a^{pred})^2$
9:     $\theta \leftarrow \theta - \alpha_\theta \nabla_\theta \mathcal{L}_{MSE}$
---

### A.2  EFFICIENT ADAPTION WITH EXPERT ACTIONS (META-IL)

With expert actions, the transformer model does not need to conduct online rollouts and updates by fine-tuning with the expert actions.

---
**Algorithm 4** Efficient Policy Adaptation with expert actions (**meta-IL**)
---
1: **Input:** testing task $\mathcal{T} \in \boldsymbol{T}^{test}$, $\text{HDT}_\psi$ with adapter parameters $\psi$, one-shot demonstration with actions $\mathcal{P}$, batch size $M$, learning rate $\alpha_\psi$, iteration $N$
2: **Initialize** adapter parameters $\psi$ with trained hyper-network
3: **for** $n = 1$ **to** $N$ **do**
4:     Sample $M$ segments $\tau$ of length $K$ from $\mathcal{P}$, and a demo. $h^o$ by omitting the actions
5:     Get a batch $\mathcal{B} = \{(h^o, \tau_m)\}_{m=1}^M$
6:     $a^{pred} \leftarrow HDT_\psi(h^o, \tau_m), \forall(h_i^o, \tau_m) \in \mathcal{B}$
7:     $\psi \leftarrow \psi - \alpha_\psi \nabla_\psi \frac{1}{|\mathcal{B}|}\sum_{(h_i^o,\tau_m)\in\mathcal{B}}(a - a^{pred})^2$
---

### A.3  HDT-IA3

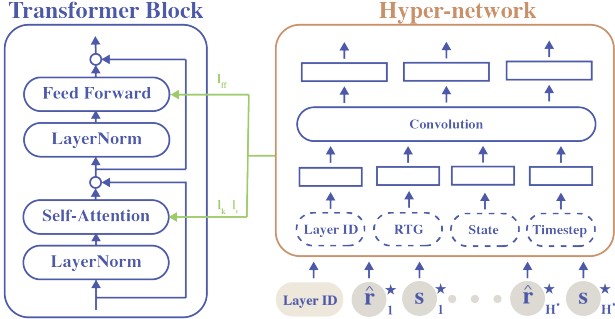

Figure 5: Model Architecture of HDT-IA3.

We show the model architecture of HDT-IA3 in Figure 5. Compared with our proposed HDT, HDT-IA3 has a hyper-network that takes the same encoding of demonstrations as HDT but outputs IA3

parameters instead. The parameter efficient fine-tuning method, IA3, proposed by (Liu et al., 2022), adds position-wise scaling weights to the self-attention and activations between feedforward layers. Considering that the transformer block in DT only contains one feedforward layer, we rescale the outputs of the feedforward layer instead. Moreover, to make a fair comparison, we use one set of IA3 parameters for all positions, similar to the position-agnostic rescaling of HDT.

Formally, the IA3 parameters include $l_k \in \mathbb{R}^{d_x}, l_v \in \mathbb{R}^{d_x}, l_{ff} \in \mathbb{R}^{d_x}$. The key $K$ and the value $v$ of the self-attention module are rescaled by $l_k$ and $l_v$, respectively. The feedforward output is rescaled by $l_{ff}$.

### A.4 MODEL SIZES

In this section, we describe the model size of different methods to help compare parameter efficiency.

Table 3: Detailed Model sizes for experiments with a large base DT model.

|  | Base model | Hypernet | Adaptation | fine-tune type | Percentage |
|---|---|---|---|---|---|
| HDT | 13M | 2.3M | 69K | Adapter layer parameters | 0.5% |
| HDT-IA3 | 13M | 0.2M | 6K | IA3 parameters | 0.05% |
| PDT | 13M | - | 6K | demonstrations | 0.05% |
| DT | 13M | - | 13M | full model parameters | 100% |

Table 4: Model sizes when different base DT models and adapter's bottleneck sizes.

|  | Embedding | blocks | heads | bottleneck size | total | DT | updated |
|---|---|---|---|---|---|---|---|
| HDT | 512 | 4 | 8 | 16 | 15.5M | 13.1M | 69K |
| HDT-medium | 128 | 6 | 4 | 16 | 2.0M | 1.3M | 26K |
| HDT-small | 128 | 3 | 1 | 16 | 1.4M | 0.7M | 13K |
| HDT-hidden-8 | 512 | 4 | 8 | 8 | 14.4M | 13.1M | 37K |
| HDT-hidden-4 | 512 | 4 | 8 | 4 | 13.9M | 13.1M | 20K |
| HDT-hidden-2 | 512 | 4 | 8 | 2 | 13.6M | 13.1M | 12K |

### A.5 HYPER-PARAMETERS

Table 5: Hyperparameters for DT-related models

| Hyperparameters | Value |
|---|---|
| $K$ (length of context $\tau$) | 20 |
| demonstration length | 200 |
| training batch size for each task $M$ | 16 |
| pre-training iterations | 4000 |
| number of gradient updates in each iteration | 10 |
| number of evaluation episodes for each task | 10 |
| learning rate $\alpha_\theta, \alpha_\phi, \alpha_\psi,$ | 1e-4 |
| learning rate decay weight | 1e-4 |
| activation | GELU |
| online rollout budget $N_{epi}$ | 4000 |
| online rollout budget in each training iteration | 20 |
| fine-tuning iterations | 200 |
| exploration $\epsilon$ | 0.2 |
| SiMPL learning rate | 5e-5 |

## B    Additional results in Meta-World benchmark

### B.1    The effect of hyper-networks through environment Visualization

We aim to show the quality of the adapter's parameter initialization by visualizing the environment rollouts. The comparisons between our proposed HDT and the baseline HDT-Rand show that initializing the adapter layer with hyper-networks provides a strong prior helping guide online exploration.

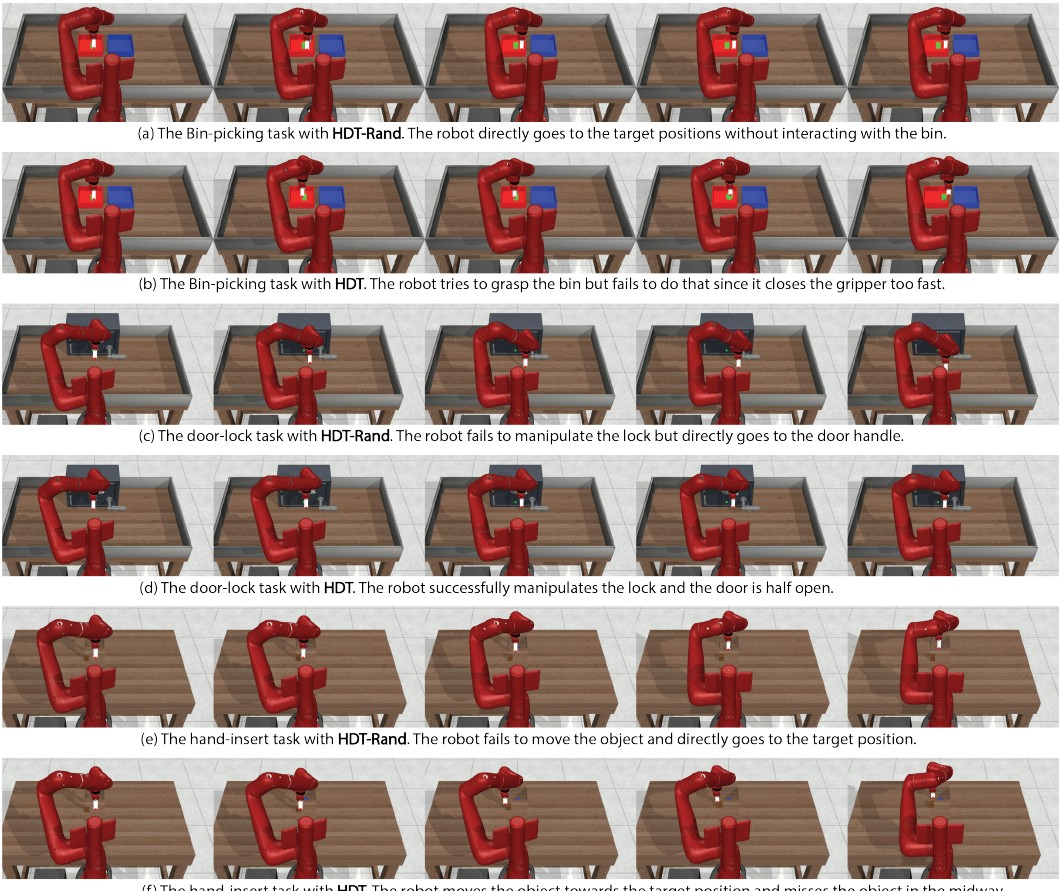

(a) The Bin-picking task with **HDT-Rand**. The robot directly goes to the target positions without interacting with the bin.

(b) The Bin-picking task with **HDT**. The robot tries to grasp the bin but fails to do that since it closes the gripper too fast.

(c) The door-lock task with **HDT-Rand**. The robot fails to manipulate the lock but directly goes to the door handle.

(d) The door-lock task with **HDT**. The robot successfully manipulates the lock and the door is half open.

(e) The hand-insert task with **HDT-Rand**. The robot fails to move the object and directly goes to the target position.

(f) The hand-insert task with **HDT**. The robot moves the object towards the target position and misses the object in the midway.

Figure 6: Environment Visualization. We provide key screenshots of three testing environment rollouts to compare the initialization of our proposed **HDT** and HDT-Rand, which randomly initialize the adapter layers.

### B.2    t-SNE visualization of adapters' parameters initialized by the hyper-network

To show whether the hyper-network extract task-specific information, we visualize the adapters' parameters initialized by the hyper-network in Figure 7. For each task, we randomly sample 100 expert demonstrations without actions, which serves as input to the hyper-network, and collect 100 samples of the adapters' parameters as the output of the hyper-network. Considering that the adapters contain around 69k parameters which form a high-dimensional vector, we first reduce the dimension to 1000 via principle component analysis (PCA). We then project the PCA results to a 2-dimensional space using t-Distributed Stochastic Neighbor Embedding (t-SNE). We use *sklearn* to conduct PCA and t-SNE. We change the perplexity to 10, considering that we use a relatively small number of samples. All the other hyperparameters except the perplexity are the same as the default parameters for PCA and t-SNE.

In Figure 7, we show the transformed 2D features for both the training and testing tasks. The training tasks are labeled with black text and circle markers. The testing tasks are labeled with red text and

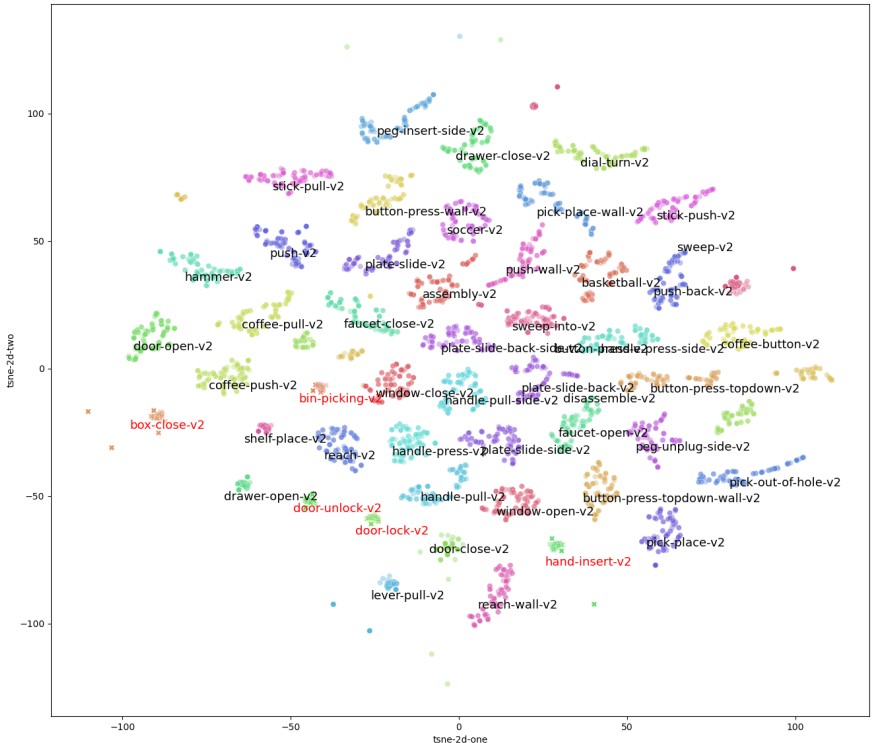

Figure 7: T-SNE visualization of the initialized adapter layers based on pre-trained hyper-network. The cross marker and red text represent testing tasks. The dot marker and black text represent training tasks.

cross markers. We can see that the adapters' parameters for different tasks could be distinguished from each other, showing that the hyper-network indeed extracts task-specific information.

## B.3 TESTING PERFORMANCES

We provide the quantitative per-task results in the 5 testing tasks in Table 6 and Table 7. More concretely, Table 6 shows the results for HDT with different base DT model sizes and adapter hidden sizes when there exist no expert actions. Table 7 shows the results for all methods when the expert actions are available.

Table 6: Per-task Quantitative results for ablation studies without expert actions (**meta-LfO**).

|  | bin-picking | | box-close | | hand-insert | | door-lock | | door-unlock | |
|  | success rate | rollouts | success rate | rollouts | success rate | rollouts | success rate | rollouts | success rate | rollouts |
|---|---|---|---|---|---|---|---|---|---|---|
| HDT-medium | $0.00 \pm 0.00$ | x | $0.00 \pm 0.00$ | x | $0.00 \pm 0.00$ | x | $0.00 \pm 0.00$ | x | $0.00 \pm 0.00$ | x |
| HDT-small | $0.00 \pm 0.00$ | x | $0.00 \pm 0.00$ | x | $0.00 \pm 0.00$ | x | $0.83 \pm 0.24$ | 30 | $0.00 \pm 0.00$ | x |
| HDT-hidden-8 | $0.25 \pm 0.43$ | 1170 | $0.13 \pm 0.22$ | 240 | $0.00 \pm 0.00$ | 3160 | $0.38 \pm 0.41$ | 40 | $0.50 \pm 0.50$ | 2140 |
| HDT-hidden-4 | $0.00 \pm 0.00$ | x | $0.25 \pm 0.43$ | 1180 | $0.00 \pm 0.00$ | x | $0.00 \pm 0.00$ | x | $0.00 \pm 0.00$ | 3020 |
| HDT-hidden-2 | $0.00 \pm 0.00$ | x | $0.13 \pm 0.22$ | 3710 | $0.00 \pm 0.00$ | x | $0.00 \pm 0.00$ | x | $0.00 \pm 0.00$ | 180 |

Table 7: Per-task Quantitative results with expert actions (**meta-IL**).

|  | bin-picking | box-close | hand-insert | door-lock | door-unlock |
|---|---|---|---|---|---|
| HDT | $1.00 \pm 0.00$ | $0.63 \pm 0.41$ | $1.00 \pm 0.00$ | $1.00 \pm 0.00$ | $0.98 \pm 0.04$ |
| PDT | $0.25 \pm 0.43$ | $0.25 \pm 0.17$ | $0.00 \pm 0.00$ | $0.33 \pm 0.41$ | $0.50 \pm 0.50$ |
| DT | $0.67 \pm 0.47$ | $0.80 \pm 0.28$ | $1.00 \pm 0.00$ | $1.00 \pm 0.00$ | $1.00 \pm 0.00$ |
| HDT-IA3 | $0.00 \pm 0.00$ | $0.00 \pm 0.00$ | $0.00 \pm 0.00$ | $0.00 \pm 0.00$ | $0.50 \pm 0.12$ |
| HDT-medium | $0.67 \pm 0.47$ | $0.17 \pm 0.24$ | $1.00 \pm 0.00$ | $1.00 \pm 0.00$ | $1.00 \pm 0.00$ |
| HDT-small | $0.16 \pm 0.24$ | $0.29 \pm 0.24$ | $0.00 \pm 0.00$ | $0.83 \pm 0.24$ | $0.17 \pm 0.24$ |
| HDT-hidden-8 | $1.00 \pm 0.00$ | $0.28 \pm 0.22$ | $1.00 \pm 0.25$ | $0.00 \pm 0.00$ | $1.00 \pm 0.00$ |
| HDT-hidden-4 | $0.00 \pm 0.00$ | $0.38 \pm 0.31$ | $0.84 \pm 0.27$ | $1.00 \pm 0.00$ | $0.69 \pm 0.42$ |
| HDT-hidden-2 | $0.00 \pm 0.00$ | $0.16 \pm 0.27$ | $0.00 \pm 0.00$ | $1.00 \pm 0.00$ | $0.69 \pm 0.23$ |

## B.4 ADDITIONAL BASELINES ON META-WORLD WITH 45 TRAINING TASKS

We train baselines on the 45 training tasks of the Meta-World ML45 benchmark. We implement CQL Kumar et al. (2020), IQL Kostrikov et al. (2021), BC and TD3BC Fujimoto & Gu (2021) based on the *d3rlpy* package Takuma Seno (2021). The main results are in Figure 8. Note that CQL and IQL do not incorporate offline demonstrations when interacting with the environments. Thus it is expected that CQL and IQL would perform worse than our proposed HDT.

From Figure 8 and Table 8, we can see that in the meta-IL setting, BC and IQL could have performance improvement but still underperform HDT. In the meta-LfO setting, CQL and IQL are trained in an online manner and underperform HDT. It is also worth noting that all four baselines here will suffer from the forgetting problem since all the model parameters are modified.

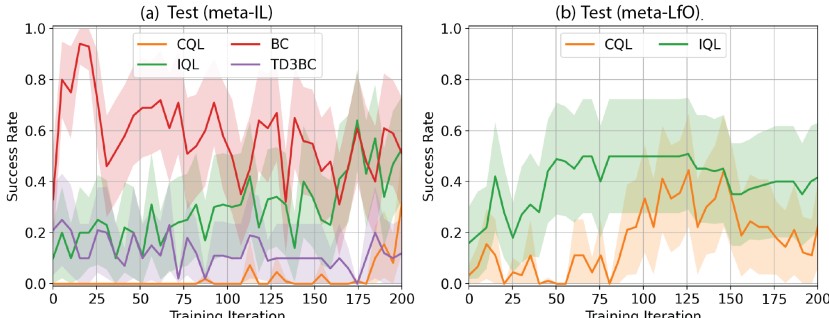

Figure 8: Qualitative results in Meta-World ML45 benchmark

Table 8: Quantitative results of baselines on Meta-World ML45 benchmarks.

|  | Baselines in Meta-World ML45 | | | |
|---|---|---|---|---|
|  | Train | Test (no-FT) | Test (meta-IL, 1 shot) | Test (meta-LfO) |
| HDT | $\mathbf{0.89 \pm 0.00}$ | $0.12 \pm 0.01$ | $\mathbf{0.93 \pm 0.10}$ | $\mathbf{0.80 \pm 0.16}$ |
| BC | $0.70 \pm 0.01$ | $0.10 \pm 0.06$ | $0.51 \pm 0.22$ | - |
| TD3+BC | $0.53 \pm 0.00$ | $0.00 \pm 0.00$ | $0.12 \pm 0.13$ | - |
| CQL | $0.06 \pm 0.01$ | $0.04 \pm 0.07$ | $0.32 \pm 0.19$ | $0.53 \pm 0.21$ |
| IQL | $0.62 \pm 0.00$ | $\mathbf{0.13 \pm 0.18}$ | $0.53 \pm 0.20$ | $0.31 \pm 0.17$ |

## B.5 ADDITIONAL EXPERIMENTS ON D4RL POINTMAZE ENVIRONMENTS

We hope to show the performance of HDT in another domain beyond manipulation tasks and thus add another set of locomotion tasks based on D4RL's pointmaze environment Fu et al. (2020). We train all the methods with 50 training tasks and test in 10 testing tasks. The detailed description of the environment and tasks are deferred to Section C.2.

The main results for the pointmaze environment are in Figure 9 and Table 9. We can see that HDT still outperforms baselines in terms of adaptation capability to unseen tasks in both meta-IL and meta-LfO settings. The results in the navigation domain help validate the main results presented in the original submission.

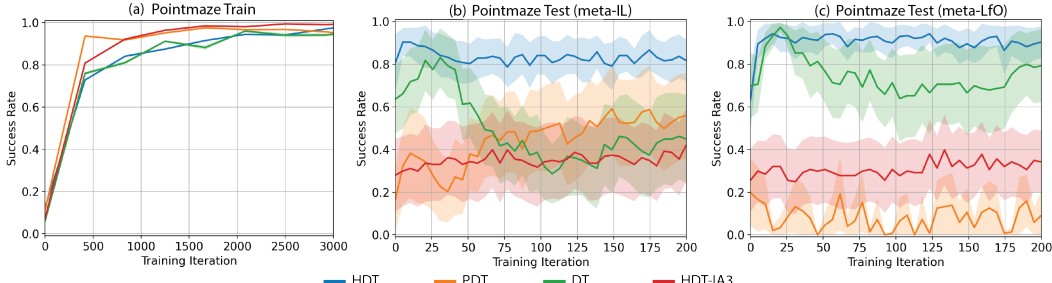

Figure 9: Qualitative results in Pointmaze benchmark

Table 9: Quantitative results on Pointmaze environments.

| | Pointmaze Peformances | | | |
| | Train | Test (no-FT) | Test (meta-IL, 1 shot) | Test (meta-LfO) |
| --- | --- | --- | --- | --- |
| HDT | **0.97 ± 0.00** | **0.73 ± 0.00** | **0.83 ± 0.10** | **0.89 ± 0.08** |
| PDT | 0.95 ± 0.00 | 0.63 ± 0.01 | 0.54 ± 0.20 | 0.11 ± 0.10 |
| DT | 0.94 ± 0.00 | 0.51 ± 0.03 | 0.46 ± 0.21 | 0.79 ± 0.16 |
| HDT-IA3 | **0.97 ± 0.00** | 0.55 ± 0.00 | 0.37 ± 0.16 | 0.33 ± 0.15 |

## C  ENVIRONMENT DETAILS

### C.1  META-WORLD EXPERIMENTS

#### C.1.1  SUCCESS CRITERION

We utilize the success signal feedback from the environment to calculate the success score, which is the sum of the success signals within one episode. To avoid false positive success identifications, we define a successful episode if the success score is larger than 5 and the total episode return is larger than 300.

### C.2  POINTMAZE EXPERIMENTS

#### C.2.1  DATA GENERATION

For each task in the training set, we use the expert rule-based controller provided in the D4RL package to generate 100 episodes. We use the medium-maze as the maze layout. For each testing task, we also use the expert controller provided but only generate one episode.

#### C.2.2  SUCCESS CRITERION

To avoid false positive success identifications, we define a successful episode if the accumulated sparse reward is larger than 5.

#### C.2.3  TRAINING AND TESTING TASKS

Each task corresponds to different start and goal positions.

The list of 50 training tasks:

- Task index: 0, start: [3,3], goal: [1,1]
- Task index: 1, start: [3,3], goal: [1,2]
- Task index: 2, start: [3,3], goal: [1,5]
- Task index: 3, start: [3,3], goal: [1,6]
- Task index: 4, start: [3,3], goal: [2,1]
- Task index: 5, start: [3,3], goal: [2,2]
- Task index: 6, start: [3,3], goal: [2,4]

- Task index: 7, start: [3,3], goal: [2,5]
- Task index: 8, start: [3,3], goal: [2,6]
- Task index: 9, start: [3,3], goal: [3,2]
- Task index: 10, start: [3,3], goal: [3,4]
- Task index: 11, start: [3,3], goal: [4,1]
- Task index: 12, start: [3,3], goal: [4,2]
- Task index: 13, start: [3,3], goal: [4,4]

- Task index: 14, start: [3,3], goal: [4,5]
- Task index: 15, start: [3,3], goal: [4,6]
- Task index: 16, start: [3,3], goal: [5,1]
- Task index: 17, start: [3,3], goal: [5,3]
- Task index: 18, start: [3,3], goal: [5,4]
- Task index: 19, start: [3,3], goal: [5,6]
- Task index: 20, start: [3,3], goal: [6,1]
- Task index: 21, start: [3,3], goal: [6,2]
- Task index: 22, start: [3,3], goal: [6,3]
- Task index: 23, start: [3,3], goal: [6,5]
- Task index: 24, start: [3,3], goal: [6,6]
- Task index: 25, start: [4,4], goal: [1,1]
- Task index: 26, start: [4,4], goal: [1,2]
- Task index: 27, start: [4,4], goal: [1,5]
- Task index: 28, start: [4,4], goal: [1,6]
- Task index: 29, start: [4,4], goal: [2,1]
- Task index: 30, start: [4,4], goal: [2,2]
- Task index: 31, start: [4,4], goal: [2,4]

- Task index: 32, start: [4,4], goal: [2,5]
- Task index: 33, start: [4,4], goal: [2,6]
- Task index: 34, start: [4,4], goal: [3,2]
- Task index: 35, start: [4,4], goal: [3,3]
- Task index: 36, start: [4,4], goal: [3,4]
- Task index: 37, start: [4,4], goal: [4,1]
- Task index: 38, start: [4,4], goal: [4,2]
- Task index: 39, start: [4,4], goal: [4,5]
- Task index: 40, start: [4,4], goal: [4,6]
- Task index: 41, start: [4,4], goal: [5,1]
- Task index: 42, start: [4,4], goal: [5,3]
- Task index: 43, start: [4,4], goal: [5,4]
- Task index: 44, start: [4,4], goal: [5,6]
- Task index: 45, start: [4,4], goal: [6,1]
- Task index: 46, start: [4,4], goal: [6,2]
- Task index: 47, start: [4,4], goal: [6,3]
- Task index: 48, start: [4,4], goal: [6,5]
- Task index: 49, start: [4,4], goal: [6,6]

The list of 10 testing tasks:

- Task index: 50, start: [3,2], goal: [1,1]
- Task index: 51, start: [3,2], goal: [4,4]
- Task index: 52, start: [3,2], goal: [2,5]
- Task index: 53, start: [3,4], goal: [4,6]
- Task index: 54, start: [3,4], goal: [1,6]

- Task index: 55, start: [3,4], goal: [2,2]
- Task index: 56, start: [4,5], goal: [6,5]
- Task index: 57, start: [4,5], goal: [3,2]
- Task index: 58, start: [4,5], goal: [5,3]
- Task index: 59, start: [5,4], goal: [3,2]

