# OpenReview forum: "Hyper-Decision Transformer for Efficient Online Policy Adaptation"
_ICLR.cc/2023/Conference — ICLR 2023 poster_

### Official Review · Reviewer_k7sT · 2022-10-13

**Confidence:** 3
**Correctness:** 3
**Technical Novelty And Significance:** 2
**Empirical Novelty And Significance:** 2
**Recommendation:** 6

**Clarity, Quality, Novelty And Reproducibility:**

See also above.

Re. correctness: The evidence is mostly empirical; I did not check all details of the experiments. Overall, the experimental section is sensibly structured (several baselines, questions answered by comparison with other models, etc.); however, in the specifics, I did not find all details understandable, e.g., the point above (re. 5.3).

**Strength And Weaknesses:**

**Strengths:**

The overall problem - multi-task learning - and the choice of the interesting recent DT architecture as base, are well motivated.

The method combines several paradigms and seems to transfer insights from NLP Transformer models, which is nice.

The experiments seem to indicate, that the method works comparably well.


**Weaknesses and specific points for improvement:**

Many paradigmas are put together (meta, offline RL, IL, …), sometimes with more, sometimes with less motivation. For instance, why combine offline RL base-training with expert-data as task-specific info? Where does this scenario happen in practice (i.e., where do we have the base offline RL data plus a demonstrator for each task)? It is not entirely counterintuitive, but I didn't see the clear motivation. Maybe the main reason is that only this setting (i.e., data from a demonstrator in the new task) fits well into the quite restrictive DT format where the data has to be in the form of s/a/r-triplets. But how about just using offline RL data for the new task as well (or is there too little information content in non-expert data?). A stronger motivation would help here.

The wiriting clarity and understanding of the method needs to be improved, e.g.,
* The action variable $a$ in Alg 1/2 comes from nowhere. It needs to be related to the trajectories etc. Essentially, variables always have to be properly introduced (within a context/algo).
* I couldn't follow the argument in sec 5.3. Why again does Fig 3 (a) show that the hyper net in fact learns to use task-specific knowledge?
* Imo, DT is introduced a bit too briefly although it is at the very core and a quite new paradigm. I didn't understand how the reward-to-go is obatained but I guess this simply the retrospective sum since we are in an offline setting - OK but maybe clarify. How about this length K "lag" fed into the transofmer. Why not, as in classic RL, just feed the *single* current state (plus desired reward in the sense of the DT idea) and output an action? Is this to account for non-Markov, or for an agent memory, or for task info? There is an informal comment on this on p4, but it feels vauge - then again, maybe one just has to accept that these sort of approaches are less motivated by understanding and more by experiments.

Further comment on experiment: I'm a bit surprised about the very poor performance of some of the baslines in Table 2 (0 success). This can be a hint that they are not tuned properly enough.

In terms of novelty/significance, there seems to be some novelty, however, the paper is heavily based on "just" combining several well-established ideas.

**Summary Of The Paper:**

As I understand: For learning an agent across multiple RL tasks that can quickly adapt to new tasks, the authors combine several established paradigmas:
- decision transformers as the base archictecture for the learning agent, trained from offline RL data across tasks;
- adding additional "adaption" parameters (layers);
- these parameters are pre-trained using the Hyper net training (Alg 1) and then used as initalization;
- when a new task arrives, with demonstrator data, then first the adaptation layer is fine-tuned (from the above initializatin) on that new task's demonstrator data (while the base params are kept). This is a trade-off to be able to be adapt, while avoiding too much variance of the overall architecture upon new tasks (and forgetting old ones).

Some of these ideas come from NLP and are here tansferred to DT.

In the experiments, HDT outperforms some baselines and some questions are tried to be answered experimentally, e.g., if the hyper net learns to use the task-specific data (5.3).

**Summary Of The Review:**

The multi-task problem is well-motivated, the interesting recent DT approach is combined with further ideas to address it; the novelty exists but is limited and the writing and explanations need to be improved. Hence, I do not see a fundamental issue; however, the contribution seems rather limited.

---

> ### Author Response · Authors · 2022-11-18
> **Response to Reviewer k7sT**
>
>
> We thank the reviewer for the constructive review and for finding our problem formulation "well-motivated". We provide a detailed response to your concerns below.
>
> 1. **A stronger motivation would help**
> - We thank the reviewer for the suggestion. We are interested in the pretraining and few-shot adaptation paradigm in the robotics domain. We are motivated to study this paradigm for mainly two reasons. First, great efforts [Ebert et al., 2021, Jang et al., 2022, Fu et al., 2022] are devoted to generating datasets for robotics that have the potential to benefit multiple robotics control tasks, like ImageNet for computer vision. Most of the datasets contain states and actions considering the sequential nature of robotics applications, and thus we hope to propose an algorithm that could benefit from the growing datasets. Second, human demonstrations are, in general, easier to acquire in realistic situations, such as structured industrial and household scenes. However, such demonstrations are usually limited due to the expensive human labor and may not necessarily contain annotated actions, such as when learning from video demonstrations. Hence we are motivated to follow a setting where there exist limited available demonstrations (with or without actions) from human (near-) experts and leverage them to achieve few-shot adaptation.
>     - Ebert, Frederik, et al. "Bridge data: Boosting generalization of robotic skills with cross-domain datasets." arXiv preprint arXiv:2109.13396 (2021).
>     - Jang, Eric, et al. "Bc-z: Zero-shot task generalization with robotic imitation learning." Conference on Robot Learning. PMLR, 2022.
>     - Fu, Haoyuan, et al. "RoboTube: Learning Household Manipulation from Human Videos with Simulated Twin Environments." 6th Annual Conference on Robot Learning.
>
> 2. **Explain the action variable**
> - Thank the reviewer for the suggestion. We modify Algorithm 1 to connect action $a$ with the demonstration dataset $\mathcal{D}$ and Algorithm 2 to connect action $a$ with the online collected data buffer. Modifications are in blue.
>
> 3. **Clarification about Section 5.3: "Why again does Fig 3 (a) show that the hyper net in fact, learns to use task-specific knowledge?"**
>  - We add another visualization of the adapter's parameters initialized by the hyper-network for each task to show whether the hyper-network extracts task-relevant information. The figure in this [link](https://i.imgur.com/TB9atDs.png) shows that the adapters' parameters for different tasks could be distinguished from each other, indicating that the hyper-network indeed extracts task-specific information. We modify Section 5.3 accordingly and add Section B.4 to present the visualization process.
>
> 4. **Clarify how to get reward-to-go, the K-lag in DT**
>  - We thank the reviewer for the suggestions. We modify Section 3.2 accordingly by introducing the reward-to-go tokens, and the benefit of using history input with length K. We agree that the benefit of using history input is mainly validated by empirical results as in [Mandlekar et al., 2021]. In our setting, where the object type and shape are not included in the state observation, historical interactions help reveal information about the object. We agree that understanding the benefit of the history input and the transformer architecture theoretically would be an exciting future direction.
>      - Mandlekar, Ajay, et al. "What matters in learning from offline human demonstrations for robot manipulation." arXiv preprint arXiv:2108.03298 (2021).
>
> 5. **The poor performance of baseline methods**
>  - To make a fair comparison, we use the same hyperparameters for all methods and try to give each method a similar (if not the same) amount of tuning. We have tried hyperparameter search for each baseline method, but we did not find significant performance improvement compared with our current hyperparameters. The performance gap shows that using gradient updates to modify the baselines methods are not effective, which motivates further carefully designed prompt tuning and parameter tuning methods for robotics tasks. We add two more sets of experiments using Meta-World ML45 benchmark (Section B.5) and pointmaze locomotion tasks (Section B.7). Both experiments help validate our argument that PDT and HDT-IA3 are, in general, hard to finetune compared with our proposed HDT.

---

### Official Review · Reviewer_aTrS · 2022-10-24

**Confidence:** 4
**Correctness:** 3
**Technical Novelty And Significance:** 3
**Empirical Novelty And Significance:** 3
**Recommendation:** 6

**Clarity, Quality, Novelty And Reproducibility:**

# Clarity
The paper is clear overall. However, it is missing some important details as explained above.

# Quality
The paper is well-motivated, proposes a sensible approach, and compares it with a number of different baselines and ablations on one domain, demonstrating promising results. However, I believe the empirical evaluation could be further improved by including evaluations on well-established benchmarks, other domains, and a few more relevant baselines, as detailed above. The paper could also benefit from more extensive analysis.

# Novelty
While the different parts of the approach aren't novel, this particular combination and application is, so I believe the work contains enough novelty for this to not be a factor against acceptance, assuming all the other more important issues are adequately addressed.

# Reproducibility
I couldn't find any mention of the code. While the method contains some details regarding the implementation and experimental setup, these are far from enough to easily reproduce the work. Do you have any plans of open sourcing the code? This is an important factor in my decision.

**Strength And Weaknesses:**

# Strengths
- important problem of interests for the community
- promising initial results, the proposed approach seems to be quite effective
- simple, intuitive approach
- some interesting insights on how the method scales with the number of training tasks and model size

# Weaknesses
### 1. evaluation on a single domain
- The method is evaluated only on the tasks from Meta World, a robotic manipulation domain. Hence, it is difficult to judge whether the results will generalize to other domains. I strongly recommend running experiments on a different benchmark such as Atari which is commonly used in the literature. This would also verify whether the method works with discrete action spaces and high-dimensional observations.

### 2. evaluation on a setting created by the authors, no well-established external benchmark
- The authors seem to create their own train and test splits in Meta World. This seems strange since Meta World recommends a particular train and test split (e.g. MT10 or MT50) in order to ensure fair comparison across different papers. I strongly suggest running experiments on a pre-established setting so that your results can easily be compared with prior work (without having to re-implement or re-run them). You don't need to get SOTA results, just show how it compares with reasonable baselines like the ones you already include. Otherwise, there is a big question mark around why you created your own "benchmark" when a very similar one exists already and whether this was somehow carefully designed to make your approach look better.

### 3. limited number of baselines
- While you do have some transformer-based baselines I believe the method could greatly benefit from additional ones like BC, transformer-BC, and other offline RL methods like CQL or IQL. Such comparisons could help shed more light into whether the transformer architecture is crucial, the hypernetwork initialization, the adaptation layers, or the training objective.

### 4. more analysis is needed
- It isn't clear how the methods compare with the given expert demonstrations on the new tasks. Do they learn to imitate the policy or do they learn a better policy than the given demonstration? I suggest comparing with the performance of the demonstration or policy from which the demonstration was collected.

- If the environment is deterministic and the agent gets to see expert demonstrations, isn't the problem of learning to imitate it quite easy? What happens if there is more stochasticity in the environments or the given demonstration isn't optimal?

- When finetuning transformers, it is often the case that they forget the tasks they were trained on. It would be valuable to show the performance of your different methods on the tasks they were trained on after being finetuned on the downstream tasks. Are some of them better than the others at preserving previously learned skills?

### 5. missing some important details
- The paper seems to be missing some important details regarding the experimental setup. For example, it wasn't clear to me how the learning from observations setting works. At some point you mention that you condition on the expert observations while collecting online data. Does this assume the ability to reset the environment in any state / observation? If so, this is a big assumption that should be more clearly emphasized and discussed. how exactly are you using the expert observations in combination with online learning?

- There are also some missing details regarding the expertise of the demonstrations at test time. Are these demonstrations coming from an an expert or how good are they?

# Minor
- sometimes you refer to generalization to new tasks. however, you finetune your models, so i believe a better term would be transfer or adaptation to new tasks.




**Summary Of The Paper:**

This paper proposes a new approach

Decision Transformers (DT) have demonstrated strong performances in offline reinforcement learning settings, but quickly adapting to unseen novel tasks remains
challenging. To address this challenge, we propose a new framework, called
Hyper-Decision Transformer (HDT), that can generalize to novel tasks from a
handful of demonstrations in a data- and parameter-efficient manner. To achieve
such a goal, we propose to augment the base DT with an adaptation module,
whose parameters are initialized by a hyper-network. When encountering unseen tasks, the hyper-network takes a handful of demonstrations as inputs and
initializes the adaptation module accordingly. This initialization enables HDT to
efficiently adapt to novel tasks by only fine-tuning the adaptation module. We validate HDT’s generalization capability on object manipulation tasks. We find that
with a single expert demonstration and fine-tuning only 0.5% of DT parameters,
HDT adapts faster to unseen tasks than fine-tuning the whole DT model. Finally,
we explore a more challenging setting where expert actions are not available, and
we show that HDT outperforms state-of-the-art baselines in terms of task success
rates by a large margin. Demos are available on our project page.

**Summary Of The Review:**

This paper proposes a new approach for finetuning on new tasks after offline training on a set of different tasks. While the authors present some promising initial results, the empirical evaluation requires more work to warrant acceptance at the ICLR conference. In particular, the generality of the approach cannot be assessed without evaluating it on multiple well-established benchmarks designed in prior work (rather than by the authors themselves). More baselines and analysis could also further strengthen the paper.

---

> ### Author Response · Authors · 2022-11-18
> **Response to Reviewer aTrS (Part 1)**
>
>
> We thank the reviewer for finding our work "well-motivated" and all the critical comments for helping improve the paper. We provide detailed responses to your questions below.
>
> 1. **Evaluation on a single domain**
> - In this paper, we aim to go beyond game settings and work on realistic robotic applications with continuous action spaces, such as manipulation tasks. It is worth noting that our tasks involve fine-grind gripper control, which casts great challenges in solving new tasks with different object shapes. Motivated by the great opportunity of real-world robot applications, including automated robot storage systems and existing challenges to handle multiple objects, we aim to focus on few-shot adaptation with a demonstration provided by experts in the robotics domain. We agree that Atari is a highly valuable benchmark with high-dimensional image inputs. However, there is no commonly-used dataset or benchmarks for evaluating the few-shot or meta-learning in the Atari domain. Hence it is not aligned with the goal of this paper, which is enhancing the few-shot generalization capability.
> - To resolve your concern, we added another set of experiments based on the widely used D4RL pointmaze navigation environment. We show that in the pointmaze experiments, HDT still outperforms baselines in both meta-IL and meta-LfO (without expert actions) settings in terms of adaptation efficiency. We add the pointmaze experiments in Section B.7. We show the main quantitative results as follows.
>
> Table A. Quantitative results in Pointmaze experiments.
> | Method | Test (meta-IL) | Test (meta-LfO) |
> |:--------: |:--------:| :--------:|
> | HDT      |  **0.83 $\pm$ 0.10**   | **0.89 $\pm$ 0.08** |
> | PDT     |   0.54 $\pm$ 0.20   | 0.11 $\pm$ 0.10 |
> | DT     |   0.46 $\pm$ 0.21   | 0.79 $\pm$ 0.16 |
> | HDT-IA3     |  0.37 $\pm$ 0.16   |  0.33 $\pm$ 0.15 |
>
> 2. **Evaluation on a setting created by the authors, no well-established external benchmark**
>  - In our original submission, we use 43 training tasks which form a subset of the original training set of the ML45 benchmark. Two tasks were omitted due to the unsatisfactory performance of the script expert policy. We want to mention that in our original experiments, the 5 testing tasks are the same as the 5 testing tasks in the Meta-World meta-learning 45 (ML45) benchmark. In our original draft, with a smaller training set, our experiment setting is actually harder than the original ML45 benchmark since we will leverage less available prior data to achieve strong adaptation capability to unseen testing tasks. In the ablation study, which has a smaller training task set (Section 5.4), we use 10 training tasks, the same as the training tasks in the Meta-World meta-learning 10 (ML10) benchmark. We modify Section C accordingly to address the similarities with prior benchmarks.
> - We did not use the multi-task learning benchmarks MT10 and MT50 since they are mainly designed for multi-task learning where the training set and test set are identical, whereas we are interested in the few-shot adaptation capability, which makes the ML10 and ML45 benchmarks are more suitable.
> - We hope to clarify that **we did not handcraft the benchmark and did largely leverage the existing Meta-World meta-learning benchmarks**. We did not cherry-pick the results since the experiment settings in this paper are harder than (if not similar to) the existing benchmark with a smaller training set, considering that the evaluation metrics are about the adaptation performance in the testing environments.
>  - To resolve your concern, we add another set of experiments using all the 45 training tasks of the Meta-World ML45 benchmark and present the results in Section B.5. We show that the results in the main text are still valid based on the results in the ML45 benchmark. We present the main quantitative results in the following table.
>
> Table B. Quantitative results in Meta-World ML45 experiments.
> | Method | Test (meta-IL) | Test (meta-LfO) |
> |:--------: |:--------:| :--------:|
> | HDT      |  **0.85 $\pm$ 0.10**   | **0.82 $\pm$ 0.14** |
> | PDT     |   0.04 $\pm$ 0.07   | 0.10 $\pm$ 0.09 |
> | DT     |   **0.85 $\pm$ 0.15**   | 0.43 $\pm$ 0.21 |
> | HDT-IA3     |  0.25 $\pm$ 0.17   |  0.24 $\pm$ 0.17 |

---

> > ### Author Response · Authors · 2022-11-18
> > **Response to Reviewer aTrS (Part 2)**
> >
> > 3. **Limited number of baselines**
> >  - Thank you for the suggestion. We address your concerns about the importance of "the transformer block, the hyper-network initialization, the adaptation layers, or the training objective" one by one.
> >  - We hope to emphasize that the most relevant and state-of-the-art baseline that focuses on the quick adaptation capability by training from offline collected data is SiMPL, which is included in the original draft. **SiMPL is a meta-learning algorithm that does not use transformer blocks.** It is unclear how to replace transformers with other neural net structures while keeping the other components consistent with HDT since the adapter layers are tailored to the transformer architecture. We add CQL, IQL, BC, and TD3+BC as baselines according to the reviewer's requirement to help further show the importance of the transformer block. We hope to address that those baselines are not designed for multi-task offline RL and not for efficient adaptation. We add Section B.6 to show the results of all baselines.
> >
> > Table C. Quantitative results of baselines in Meta-World ML45 experiments.
> > | Method | Test (meta-IL) | Test (meta-LfO) |
> > |:--------: |:--------:| :--------:|
> > | HDT      |  **0.85 $\pm$ 0.10**   | **0.82 $\pm$ 0.14** |
> > | BC       |  0.51$\pm$ 0.22  |  - |
> > | TD3+BC   |  0.12$\pm$ 0.13  |  - |
> > | CQL      |  0.32$\pm$ 0.19  | 0.53$\pm$ 0.21  |
> > | IQL      |  0.53$\pm$ 0.20  | 0.31$\pm$ 0.17  |
> >
> >  - We are unsure about the meaning of the **hyper-network initialization** and look forward to further clarification.
> >      - We believe that the initialization of the hyper-network's parameters is less important in our setting, considering that the hyper-network is trained with offline data with a large number of iterations. In our implementation, we randomly initialize hyper-networks parameters. Figure 3 (a) also shows that the hyper-network training (HDT) has a small variance in terms of the training task performance.
> >      - If the reviewer refers to using hyper-network to initialize the adapter layers' parameters, we hope to clarify that this kind of initialization for the adapter layers is quite important for solving downstream tasks. The HDT-Rand baseline, which does not initialize the adapter's parameters with the pre-trained hyper-network, performs worse than HDT with the adapter initialization, as shown in Figure 4d.
> >  - We showed the importance of **the adapter layer** by comparing HDT with HDT-IA3, which is another way of injecting parameters to be updated during fine-tuning. We justify that the adapter layer is a better design choice considering that HDT-IA3 hardly improves during adaptation in both the meta-IL and meta-LfO (Figure 3 b-c) settings.
> >  - Our **training objective** (the loss function) is the same as the decision transformer paper. We also tried to use the negative log-likelihood with entropy regularization but found no improvement over the current L2 objective.

---

> > > ### Author Response · Authors · 2022-11-18
> > > **Response to Reviewer aTrS (Part 3)**
> > >
> > > 4. **More analysis is needed**
> > >
> > > > It isn't clear how the methods compare with the given expert demonstrations on the new tasks. Do they learn to imitate the policy or do they learn a better policy than the given demonstration? I suggest comparing with the performance of the demonstration or policy from which the demonstration was collected.
> > >
> > >  - We use a rule-based expert policy for collecting data for both training and testing tasks. Each task has a tailored expert script policy provided by the Meta-World benchmark, which has an average success rate of 1. Please note that the script policy we used for collecting data can be treated as an oracle. We measure the performance of different methods based on the average success rate, which is widely adopted in existing literature when using the Meta-World benchmark and also in the original Meta-World paper [Yu et al., 2020]. By collecting expert data for each task, we hope the agent to mimic the behavior of the expert policy. Larger success rates indicate closer performances to the oracle policy. We modify Section 4.1 to mention the performance of the expert policies for data collection. The modified texts are in blue.
> > >     - Yu, Tianhe, et al. "Meta-world: A benchmark and evaluation for multi-task and meta reinforcement learning." Conference on robot learning. PMLR, 2020.
> > >
> > > > If the environment is deterministic and the agent gets to see expert demonstrations, isn't the problem of learning to imitate it quite easy? What happens if there is more stochasticity in the environments or the given demonstration isn't optimal?
> > > - We would like to emphasize that we consider two experiment settings, the meta imitation learning (meta-IL, Table 1) where the demonstrations contain expert actions in the testing time, as well as the meta-learning-from-observation (meta-LfO, Table 2), where the demonstrations contain no expert actions in the testing time. **In the challenging meta-LfO setting, expert actions are not available**, making mimicking expert actions infeasible. Learning from expert observations that contain states and rewards is a nontrivial and open problem [Torabi et al., 2019, Haldar et al., 2022]. We empirically show that our proposed hyper-DT could adapt to unseen tasks in the meta-LfO setting without using expert actions and outperforms baselines by a large margin (please refer to Table 2 and Figure 3c).
> > >     - Torabi, Faraz, Garrett Warnell, and Peter Stone. "Recent advances in imitation learning from observation." arXiv preprint arXiv:1905.13566 (2019).
> > >     - Haldar, Siddhant, et al. "Watch and match: Supercharging imitation with regularized optimal transport." arXiv preprint arXiv:2206.15469 (2022).
> > >
> > > > When finetuning transformers, it is often the case that they forget the tasks they were trained on. It would be valuable to show the performance of your different methods on the tasks they were trained on after being finetuned on the downstream tasks. Are some of them better than the others at preserving previously learned skills?
> > > - We thank the reviewer for raising the question. Please note that during adaptation, HDT only finetunes adapters, which are initialized by the pre-trained hyper-network, and leaves the pertained hyper-network’s and base DT’s parameters frozen. Hence, given the same demonstration for one training task, the hyper-network will generate the same adapter layers inserted in the transformer block considering the hyper-network’s parameters are frozen, which leads to unsacrificed performance for training tasks. Hence, **HDT’s adaptation during testing time does not affect its performance in training tasks and thus does not suffer from the so-called forgetting problem.**

---

> > > > ### Author Response · Authors · 2022-11-18
> > > > **Response to Reviewer aTrS (Part 4)**
> > > >
> > > > 5. **Missing some important details**
> > > >
> > > > > The paper seems to be missing some important details regarding the experimental setup. For example, it wasn't clear to me how the learning from observations setting works. At some point you mention that you condition on the expert observations while collecting online data. Does this assume the ability to reset the environment in any state / observation? If so, this is a big assumption that should be more clearly emphasized and discussed. how exactly are you using the expert observations in combination with online learning?
> > > >  - We would like to first clarify how expert observation is used in the online learning setting and how it helps in online learning. In the meta learning-from-observation (meta-LfO) setting, HDT initializes demonstration-conditioned adapter layers’ weights, by feeding the (state, reward-to-go, …) sequence of the demonstration to the hyper-network, which generates the adapters’ weights (as in the right most subfigure in Figure 2). The initialized adapter layer contains task-specific information extracted from the expert demonstration distilled by the hyper-network and provides a strong policy prior, as shown in ablation studies and demonstration videos on our anonymous website. With the initialized adapters’ parameters, the transformer-based policy can quickly collect successful online rollouts by interacting with the testing environment, with the number of rollouts required shown in Table 2.
> > > >  - We hope to clarify that we did not reset the environment in any state/observation, although we acknowledge that this is an effective method, as shown in [Peng et al. 2018] in a setting closely related to learning from demonstration.
> > > >      - Peng, Xue Bin, et al. "Deepmimic: Example-guided deep reinforcement learning of physics-based character skills." ACM Transactions On Graphics (TOG) 37.4 (2018): 1-14.
> > > >
> > > > > There are also some missing details regarding the expertise of the demonstrations at test time. Are these demonstrations coming from an an expert or how good are they?
> > > >  - We collect data based on the rule-based script control policies provided by the Meta-World benchmark, which have an average success rate of 1. We augment our data collection description in section 4.1 by clearly mentioning the performance of the expert script policies.
> > > >
> > > > 6. **Reproducibility**
> > > >  - Thank you for asking the question. Yes, we will open-source our code alongside the paper's camera-ready version. We hope our work could serve as a baseline in the future and help other researchers build exciting methods to achieve efficient few-shot adaptation.

---

> > > > > ### Comment · Reviewer_aTrS · 2022-11-24
> > > > > **Post-Rebuttal Response**
> > > > >
> > > > > Thank you for the detailed response, additional experiments, and paper updates. Most of my concerns have been addressed, so I will raise my score to 6.

---

### Official Review · Reviewer_FfBQ · 2022-10-24

**Confidence:** 4
**Correctness:** 3
**Technical Novelty And Significance:** 4
**Empirical Novelty And Significance:** 4
**Recommendation:** 8

**Clarity, Quality, Novelty And Reproducibility:**

The paper is well-written and easy to read. I like the demo webpage with a clear summary of main ideas and performance.

**Strength And Weaknesses:**

Strength:
The problem is well motivated, and efficiently adapting large pertained decision transformers with a handful of observation is an important line of research. Also the LfO setting is interesting since in many real life robotic applications it’s hard to obtain state action pairs as demonstrations. The paper is also easy to follow.

What I like most is the experiment design and the details. While at first sight, the proposed method seems very complicated, the design is justified in the ablation study. E.g., Fig 3 suggests that the hypernet is better than other simpler ways of adapting transformers like prompting (PDT) and IA3. It’s also nice to see the comparison with meta RL methods like SimPL, but it’s expected that SimPL is not as good as the proposed method, since SimPL does not take into the demonstration in the unseen tasks. It’s also interesting to see that the hyper net can encode the task-specific information as in sec5.3. While the paper only has one environment Metaworld, but meta45 is a pretty challenging task alone. I think this could be interesting for researchers studying decision transformers.

Weakness:
While I think metaworld is enough, more environments could be nicer, like Kitchen, which seems to be more widely adopted in the previous literature.


**Summary Of The Paper:**

In this paper, authors study the important question of efficiently adapting decision transformers when having access to a handful of demonstrations for unseen tasks. They propose to augment the base DT with an adaptation module that can be controlled by a hyper-net, which can take the demonstration as inputs. The author conduct experiments on meta world and shows that their results are better than other baselines.

**Summary Of The Review:**

In general, I think it's a good paper and recommend acceptance.

---

> ### Author Response · Authors · 2022-11-18
> **Response to Reviewer FfBQ**
>
> Thank you for your encouraging review! We are glad that you find our paper "well-written and easy to read", and the ablation study helping justify the methodology design.
>
> 1. **Add another set of experiments**
> - We agree with the reviewer that more experiments are helpful to show the generalizability of our proposed HDT to different settings. We add one set of navigation experiments based on the D4RL pointmaze environment. We show that in the pointmaze experiments, HDT still outperforms baselines in both meta-IL and meta-LfO (without expert actions) settings. We believe the new set of experiments helps strengthen the paper. We present the new set of experiments and experiment details in Section B.7 and the main quantitative results in the following table.
>
> Table A. Quantitative results in Pointmaze experiments.
> | Method | Test (meta-IL) | Test (meta-LfO) |
> |:--------: |:--------:| :--------:|
> | HDT      |  **0.83 $\pm$ 0.10**   | **0.89 $\pm$ 0.08** |
> | PDT     |   0.54 $\pm$ 0.20   | 0.11 $\pm$ 0.10 |
> | DT     |   0.46 $\pm$ 0.21   | 0.79 $\pm$ 0.16 |
> | HDT-IA3     |  0.37 $\pm$ 0.16   |  0.33 $\pm$ 0.15 |

---

### Official Review · Reviewer_9MRj · 2022-10-25

**Confidence:** 4
**Correctness:** 3
**Technical Novelty And Significance:** 3
**Empirical Novelty And Significance:** 3
**Recommendation:** 8

**Clarity, Quality, Novelty And Reproducibility:**

Except for some parts, this paper is well-written and clearly explains and evaluates their model. Their model is novel, different from Prompt-DT or IA3 methods and outperforms them. Through the details about the hyperparameters, it looks reproducible.

**Strength And Weaknesses:**

Strength
- This paper proposes a new way for DT to adapt the unseen tasks through additional neural network layers called an adapter layer. It shows better performance than baselines and model variations.
- Through designing, their adaptation can cover not just full trajectories but also demonstration data. Their model can adapt the unseen tasks with and without expert action.
- Their model is compared with many variations, such as without hyper-network initialization or training the adapter with DT simultaneously.

Weaknesses
- Some parts of this paper are not clear to me.
    - In section 3.4 equation (3), the convolutional network's input is unclear. Why did you concatenate $L_s(h^o)+L_t(h^o)+L_l(h^o)$ and $L_{\hat{r}}(h^o)+L_t(h^o)+L_l(h^o)$? Is it possible to use $L_s(h^o)+L_t(h^o)+L_l(h^o)+L_{\hat{r}}(h^o)$?
    - In section 4.2 HDB-IA3, what is the meaning that the models do not utilize position-wise rescaling?
    - Table 2 is hard to understand. Does that mean the success rate? The average success rate when achieving to get a success episode? You mentioned that the data efficiency is based on the top-2 number of online rollout episodes until a success episode. Does it mean the number in the table is the average of the two number of online rollout episodes?
    - In section 5.2, relating to the unclarity of table 2, could you explain more about why PDT hardly improves success rates?
    - > First, HDT achieves similar training task performance compared with the pre-trained DT model as in Figure 3 (a), which shows that the hyper-network could distinguish the task-specific information of training tasks.
        - Is it because the multi-task DT can distinguish the task-specific information through a given state, action, and reward sequence?
- Typo in conclusion. grinned -> grind.

**Summary Of The Paper:**

This paper proposes a new way to apply Decision-Transformer (DT) [1] to unseen tasks through the adapter layer initialized by a Hyper-network. They showed their method could adapt faster and better to unseen tasks with few parameter updates when comparing with previous works Prompt-DT [2], SiMPL [3], IA3 [4], and variations of their model. Especially their model can adapt even only with the demonstration of unseen tasks where expert action is unavailable.


**Summary Of The Review:**

This paper proposes a new method for DT to adapt the unseen tasks. The way to do this is novel and shows better performance compared with previous works [2,3,4]. For me, some parts are unclear, which I hope to be updated, but overall, this paper is well-written and good to share in our community.


[1]Chen, Lili, et al. "Decision transformer: Reinforcement learning via sequence modeling." Advances in neural information processing systems 34 (2021): 15084-15097.

[2] Xu, Mengdi, et al. "Prompting decision transformer for few-shot policy generalization." International Conference on Machine Learning. PMLR, 2022.

[3] Nam, Taewook, et al. "Skill-based Meta-Reinforcement Learning." arXiv preprint arXiv:2204.11828 (2022).

[4] Liu, Haokun, et al. "Few-shot parameter-efficient fine-tuning is better and cheaper than in-context learning." arXiv preprint arXiv:2205.05638 (2022).

---

> ### Author Response · Authors · 2022-11-18
> **Response to Reviewer 9MRj**
>
> Thank you for your constructive feedback. We are glad that you find our proposed methodology novel and "good to share" with the community. Here we address all of the issues in the weaknesses section with draft modification and new visualization.
>
> 1. **The input of the convolutional network**
>  - We thank the reviewer for raising the question. By concatenating $L_s(h^o)+L_t(h^o)+L_l(h^o)$ and $L_{\hat{r}}(h^o)+L_t(h^o)+L_l(h^o)$, which represent state and action embeddings, respectively, we rely on the convolutional neural nets to learn the weighted sum of the two modalities automatically. In contrast, adding them together would assume equal weights on $L_s$ and $L_{\hat r}$. Considering that $L_s$ and $L_{\hat r}$ correspond to different modalities, we believe that concatenating, which leads to possibly different weights, would be a better design choice. We also hope to clarify that $L_t$ and $L_l$ represent two kinds of positional embeddings, that are trajectory timestep embedding and transformer block embedding, respectively. $L_t$ and $L_l$ are shared across two different modalities, including state and rewards, and thus are added to both $L_s$ and $L_{\hat r}$.
>
> 2. **The meaning of not using the point-wise rescaling in HDT-IA3 (section 4.2)**
>  - Thank the reviewer for raising the question. In our implementation, HDT-IA3 trains hyper-network to generate rescaling weights shared by all positions in each block. We modify Section 4.2 accordingly to clarify the point further.
>
> 3. **Table 2 is hard to understand**
>  - The "success rate" column in Table 2 shows the average success rate for each testing task. The "data eff." is the average of the smallest and the second smallest number of episodes required to sample a successful episode. A smaller number corresponding to "data eff." means higher data efficiency during online rollouts. We have modified the caption of Table 2 to clarify the column's meanings.
>
> 4. **Explanation about why PDT hardly improves success rate**
>  - When fine-tuning PDT, we freeze the model parameters and update the prompt with gradient descent. The objective and hyperparameters are consistent with HDT. We observe that the updated prompt easily gets stuck in local minima. We conjecture that the gradient updates backpropagated to the prompt vanishes due to a large number of model parameters. Moreover, prompts have a limited effect on the model. They essentially affect the model output by changing the input to the model, while adapters change the model's parameters and are inserted into the middle of the transformer blocks (closer to the output end), which may relieve HDT from the gradient vanishing problem.
>
> 5. **Clarification about Sec 5.3: what is the reason that "the hyper-network could distinguish the task-specific information of training tasks"**
>  - We agree with the reviewer that, based on the training curve in Figure 3 (a), the base DT model trained with multiple tasks could distinguish the task-specific information based on the K-step historical context input, which contains a sequence of reward-to-gos, states, and actions. In Section 5.3, we aim to show whether the trained hyper-network could extract task-specific information after freezing the base DT model and purely training the hyper-network. We show that the hyper-network could indeed extract the task-relevant information by visualizing the adapter's parameters initialized by the hyper-network for each task. The figure in this [link](https://i.imgur.com/TB9atDs.png) shows that the adapters' parameters for different tasks could be distinguished from each other, indicating that the hyper-network indeed extracts task-specific information. We modified Section 5.3 accordingly and added Section B.4 to present the visualization process.

---

### Author Response · Authors · 2022-11-18
**General Response**

Dear Reviewers:

Thank you all for the constructive feedback and encouraging review. In addition to the detailed comments to each reviewer's questions, we would like to highlight our key contributions and the new experiments we added during the rebuttal phase.

**1. Our Contributions**

We are glad that you find that our work is well-motivated [FfBQ, aTrS, k7sT] focusing on an important problem with demonstrations that only contain observations [9MRj, FfBQ], our methodology is novel [9MRj, aTrS], our paper is well-written and good to share with the community [9MRj, FfBQ], and our experiments are extensively conducted with ablations and demonstrate the strong performance of our proposed HDT [9MRj, FfBQ, aTrS, k7sT]. We would like to restate our main contributions here.
 - We propose Hyper-Decision Transformer (HDT), **a novel data- and parameter-efficient transformer-based paradigm**, aiming to quickly adapt to downstream unseen tasks through pretraining on large training sets and one demonstration without actions in unseen tasks.
 - Our proposed HDT paradigm with only fine-tuning **0.5%** parameters of the original model could achieve similar asymptotic performances to fine-tuning the whole model parameter in the meta-IL setting. In the more challenging LfO setting without expert actions, HDT could quickly sample successful episodes that benefit from the strong adapter prior initialized by the trained hyper-network.
 - Our proposed HDT **does not suffer from the forgetting problem**. In other words, fine-tuning in the unseen tasks does not decrease the performance in the training tasks since HDT only fine-tunes the adapter layer inserted into the transformer blocks.

**2. New Experiments and Visualization**
 - (Reviewer 9MRj, k7sT) **Clarification about Section 5.3 related to the benefit of hyper-network:** We rewrite Section 5.3 and add a visualization of the adapter's parameters initialized by the hyper-network in Section B.4 to support that the hyper-network could distinguish task-specific information. The figure can be found in this [link](https://i.imgur.com/TB9atDs.png).
 - (Reviewer FfBQ, aTrS) **New evaluation domain:** We add another set of navigation experiments based on the D4RL benchmark and present the main results in Section B.7. The new experiments validate the main claims in our original submission.
 - (Reviewer aTrS) **Evaluate in Meta-World's benchmark:** We add another set of experiments that exactly follow the original Meta-World's ML45 benchmark. We show the results in Section B.5. Please note that the new experiment results help strengthen our original claims.
 - (Reviewer aTrS, k7sT) **Additional baselines:** We add non-transformer baselines based on `d3rlpy` package and present the results in Section B.6. Please note that the baselines still underperform our proposed HDT in terms of the adaption capability.

We have revised our paper with substantial changes to address all of your questions. Please don’t hesitate to let us know of any additional comments on the draft or the changes!

Best,

Authors

---

### Decision · Program_Chairs · 2023-01-20

**Decision:**

Accept: poster

**Justification For Why Not Higher Score:**

The writing can still be improved, and additional experiments can be beneficial in terms of more domains, more baselines, and external benchmark.

**Justification For Why Not Lower Score:**

The paper is well written and addresses an important problem with a novel technique.  Empirically, it outperforms the SOTA.

**Metareview: Summary, Strengths And Weaknesses:**

This paper enables decision transformers to rapidly adapt to novel tasks with good sample and parameter efficiency.  By employing an adaptation module with a hyper-network to initialize its parameters, only fine-tuning of the module is needed for new tasks.  Experimental results on object manipulation shows the method can adapt quite efficiently.

The paper is well written and addresses an important problem with a novel technique.  Empirically, it outperforms the SOTA.  The writing can still be improved, and additional experiments can be beneficial in terms of more domains, more baselines, and external benchmark.


**Note From Pc:**

if the above contains the word "oral" or "spotlight" please see: "oral" presentation means -> notable-top-5% and "spotlight" means -> notable-top-25%. As stated in our emails, we are disassociating presentation type from AC recommendations